# A New Fuzzy MARCOS Method for Road Traffic Risk Analysis

**Miomir Stanković [1], Željko Stević [2],\* , Dillip Kumar Das [3] , Marko Subotić [2] and Dragan Pamučar [4]**

[1]  Mathematical Institute of the Serbian Academy of Sciences and Arts, 11001 Belgrade, Serbia; miomir.stankovic@mi.sanu.ac.rs

[2]  Faculty of Transport and Traffic Engineering, University of East Sarajevo, 74000 Doboj, Bosnia and Herzegovina; marko.subotic@sf.ues.rs.ba

[3]  Civil Engineering, School of Engineering, University of Kwazulu Natal, Durban 4041, South Africa; dasd@ukzn.ac.za

[4]  Department of Logistics, University of Defence in Belgrade, 11000 Belgrade, Serbia; dragan.pamucar@va.mod.gov.rs

\*  Correspondence: zeljkostevic88@yahoo.com or zeljko.stevic@sf.ues.rs.ba

**Abstract:** In this paper, a new fuzzy multi-criteria decision-making model for traffic risk assessment was developed. A part of a main road network of 7.4 km with a total of 38 Sections was analyzed with the aim of determining the degree of risk on them. For that purpose, a fuzzy Measurement Alternatives and Ranking according to the COmpromise Solution (fuzzy MARCOS) method was developed. In addition, a new fuzzy linguistic scale quantified into triangular fuzzy numbers (TFNs) was developed. The fuzzy PIvot Pairwise RElative Criteria Importance Assessment—fuzzy PIPRECIA method—was used to determine the criteria weights on the basis of which the road network sections were evaluated. The results clearly show that there is a dominant section with the highest risk for all road participants, which requires corrective actions. In order to validate the results, a comprehensive validity test was created consisting of variations in the significance of model input parameters, testing the influence of dynamic factors—of reverse rank, and applying the fuzzy Simple Additive Weighing (fuzzy SAW) method and the fuzzy Technique for Order of Preference by Similarity to Ideal Solution (fuzzy TOPSIS). The validation test show the stability of the results obtained and the justification for the development of the proposed model.

**Keywords:** Fuzzy MARCOS; Fuzzy PIPRECIA; traffic risk; TFN; MCDM

## 1. Introduction

Multi-criteria decision-making (MCDM) methods, [1–4] especially in integration with fuzzy theory [5–7], are a very powerful and useful tool for reliable decision-making in different fields of decision-making. The decision-making process, according to Stojić et al. [8], requires the prior definition and fulfillment of certain factors, especially when it comes to solving problems in complex areas. The theory of multi-criteria decision-making according to Zavadskas et al. [9] holds a special place in the field of science. The application of fuzzy MCDM methods contributes to a more precise determination of an acceptable solution, especially since considering them with respect to different factors is a very demanding and difficult task. Moreover, this was expressed in group decision-making processes [10]. The main motivation of this study can be considered in the following way: determining of risk level at road sections requires the inclusion of a lot of different variables. After experimental data collection, variables must be assessed in a clear and precise way. For this

purpose, a new fuzzy linguistic scale was developed with quantification in TFNs. Moreover, a new fuzzy Measurement Alternatives and Ranking according to COmpromise Solution (Fuzzy MARCOS) was developed in this paper to evaluate sections of road infrastructure with the aim of determining the degree of risk on them. The development of new fuzzy MARCOS method and defining a new linguistic scale based on TFNs represent the main contributions of this paper. The advantages of the Fuzzy MARCOS method are as follows: consideration of fuzzy reference points through the fuzzy ideal and fuzzy anti-ideal solution at the very beginning of model formation, more precise determination of the degree of utility with respect to both set solutions, proposal of a new way of determining utility functions and its aggregation, possibility to consider a large set of criteria and alternatives, as demonstrated through a realistic example, too. This paper considers one of four the most important factors that affect traffic accidents and lead to hazardous situations, and it relates to the road. There are frequent situations where geometric characteristics can negatively affect the creation of potential situations and increase the risk for each traffic participant. The geometric characteristics of the road on particular sections have a major impact on increasing the risk of traffic accidents. Morency et al. [11] analyzed the extent to which road geometry and traffic volume influence social inequalities in pedestrian, cyclist and motorcyclist injuries in wealthy and poor urban areas. Based on their observational study, it was concluded that there were more injured pedestrians, cyclists and motorcyclists at intersections in poorer areas than in wealthier areas. Nevertheless, studies have shown that the two most important road factors affecting accident rates are the pavement condition and the geometric characteristics of the road [12]. Therefore, in this paper, on the basis of causal factors, the degree of risk was determined on sections of 200 m. In her research, Nenadić [13] carried out the evaluation of sections, i.e., three locations, based on seven criteria. In contrast to this research, the optimality criterion was set in such a way that the safest section was considered instead of the one with the highest risk. In the study performed by Bao et al. [14], the fuzzy methodology was used for similar purposes. An Improved hierarchical fuzzy TOPSIS model has been defined for evaluating road safety performance. The evaluation of safety performance indicators (SPIs) was performed by Khorasani et al. [15] as an MCDM problem in a methodological sense. The TOPSIS method in combination with other techniques was also used in Haghighat's research [16] for determining the safety position of the roads of the Bushehr province.

The rest of the paper is organized as follows. Section 2 provides the preliminaries necessary to develop the fuzzy MARCOS method. They refer to displaying basic operations with fuzzy numbers and presenting the steps of the crisp MARCOS method. Section 3 presents the development of the fuzzy MARCOS method algorithm (Figure 1), which consists of a total of 10 steps. In Section 4 of the paper, the MCDM model is formed and a detailed calculation of each step of the developed fuzzy MARCOS method is presented. The calculation of the criterion weight using the fuzzy PIPRECIA method is also summarized. The following Section 5 presents an overview of validation tests to verify the stability of the proposed model. Finally, Section 6 provides a brief overview of the most important tasks accomplished and the contributions of the paper along with future research guidelines.

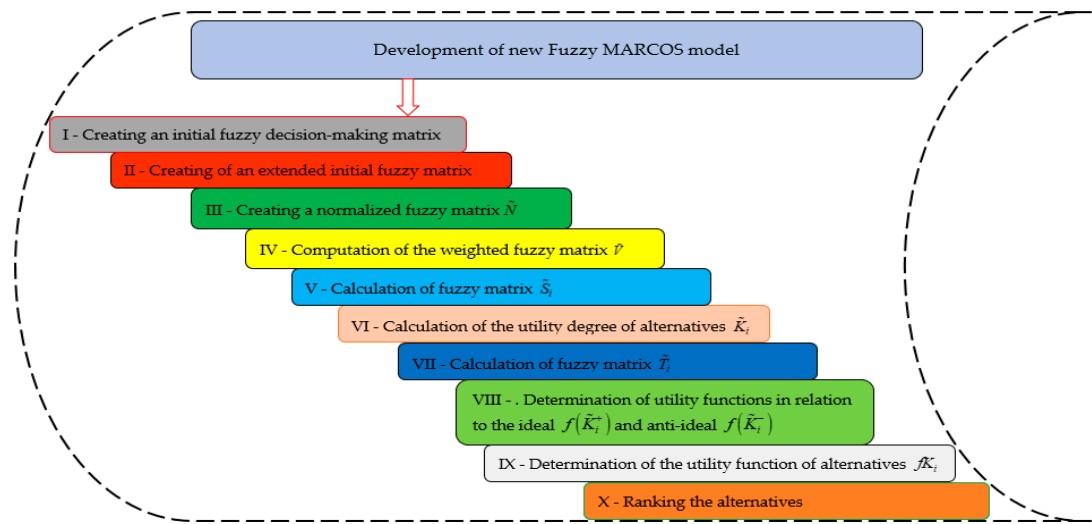

**Figure 1.** Algorithm of new developed fuzzy MARCOS method.

## 2. Preliminaries

A fuzzy number $\tilde{A}$ on R to be a TFN if its membership function $\mu_{\tilde{A}}(x)$: R→[0,1] is equal to Equation (1):

$$\mu_{\tilde{A}}(x) = \begin{cases} \frac{x-l}{m-l} & l \leq x \leq m \\ \frac{u-x}{u-m} & m \leq x \leq u \\ \\ 0 & otherwise \end{cases} \tag{1}$$

where $l$ represents the lower and $u$ upper bounds of the fuzzy number $\tilde{A}$ and $m$ is the modal value. The TFN can be marked as $\tilde{A} = (l, m, u)$.

The operations of TFN $\tilde{A}_1 = (l_1, m_1, u_1)$ and $\tilde{A}_2 = (l_2, m_2, u_2)$ are as follow [17,18]

Addition: $\tilde{A} = (l_1, m_1, u_1)$

$$\tilde{A}_1 \oplus \tilde{A}_2 = (l_1, m_1, u_1) + (l_2, m_2, u_2) = (l_1 + l_2, m_1 + m_2, u_1 + u_2). \tag{2}$$

Multiplication:

$$\tilde{A}_1 \otimes \tilde{A}_2 = (l_1, m_1, u_1) \otimes (l_2, m_2, u_2) = (l_1 \times l_2, m_1 \times m_2, u_1 \times u_2). \tag{3}$$

Subtraction:

$$\tilde{A}_1 - \tilde{A}_2 = (l_1, m_1, u_1) - (l_2, m_2, u_2) = (l_1 - u_2, m_1 - m_2, u_1 - l_2). \tag{4}$$

Division:

$$\frac{\tilde{A}_1}{\tilde{A}_2} = \frac{(l_1, m_1, u_1)}{(l_2, m_2, u_2)} = \left( \frac{l_1}{u_2}, \frac{m_1}{m_2}, \frac{u_1}{l_2} \right). \tag{5}$$

Reciprocal:

$$\tilde{A}_1^{-1} = (l_1, m_1, u_1)^{-1} = \left( \frac{1}{u_1}, \frac{1}{m_1}, \frac{1}{l_1} \right). \tag{6}$$

The following section provides a brief overview of the crisp MARCOS method defined by Stević et al. [19]:

Step 1: Designing of an initial decision-making matrix.

Step 2: Designing of an *extended* initial matrix, performed by defining the anti-ideal (*AAI*) and ideal (*AI*) solution.

$$
X = \begin{array}{c} \\ AAI \\ A_1 \\ A_2 \\ \dots \\ A_m \\ AI \end{array}
\begin{array}{c} C_1 \quad C_2 \quad \dots \quad C_n \\
\left[ \begin{array}{cccc}
x_{aa1} & x_{aa2} & \dots & x_{aan} \\
x_{11} & x_{12} & \dots & x_{1n} \\
x_{21} & x_{22} & \dots & x_{2n} \\
\dots & \dots & \dots & \dots \\
x_{m1} & x_{22} & \dots & x_{mn} \\
x_{ai1} & x_{ai2} & \dots & x_{ain}
\end{array} \right]
\end{array}
\tag{7}
$$

(AAI) is the worst alternative, while (*AI*) is best alternative. Depending on type of the criteria, *AAI* and AI are defined by applying Equations (8) and (9):

$$
AAI = \min_i x_{ij} \ if \ j \in \ B \ and \ \max_i x_{ij} \ if \ j \in C \tag{8}
$$

$$
AI = \max_i x_{ij} \ if \ j \in \ B \ and \ \min_i x_{ij} \ if \ j \in C \tag{9}
$$

*B* belongs maximization group of criteria, while *C* belongs the minimization group of criteria.

Step 3: Normalization of previous matrix (X). $N = \left[ n_{ij} \right]_{m \times n}$ are obtained using Equations (10) and (11):

$$
n_{ij} = \frac{x_{ai}}{x_{ij}} \ if \ j \in \ C \tag{10}
$$

$$
n_{ij} = \frac{x_{ij}}{x_{ai}} \ if \ j \in \ B \tag{11}
$$

where elements $x_{ij}$ and $x_{ai}$ represent the elements of the matrix *X*.

Step 4: Determination of the weighted matrix $V = \left[ v_{ij} \right]_{m \times n}$ Equation (12).

$$
v_{ij} = n_{ij} \times w_j \tag{12}
$$

Step 5: Computation of the utility degree of alternatives *Ki*—Equations (13) and (14).

$$
K_i^- = \frac{S_i}{S_{aai}} \tag{13}
$$

$$
K_i^+ = \frac{S_i}{S_{ai}} \tag{14}
$$

where $S_i$ (*I* = 1, 2, . . . , *m*) represents the sum of the elements of matrix *V*, Equation (15).

$$
S_i = \sum_{i=1}^{n} v_{ij} \tag{15}
$$

Step 6: Determination of the utility function of alternatives $f(K_i)$ defined by Equation (16).

$$
f(K_i) = \frac{K_i^+ + K_i^-}{1 + \frac{1 - f(K_i^+)}{f(K_i^+)} + \frac{1 - f(K_i^-)}{f(K_i^-)}}; \tag{16}
$$

where $f\left(K_i^-\right)$ represents the utility function in relation to (*AAI*), while $f\left(K_i^+\right)$ represents the utility function in relation to the (*AI*).

Utility functions in relation to (*AI*), and (*AAI*), solution were determined using Equations (17) and (18).

$$f\left(K_i^-\right) = \frac{K_i^+}{K_i^+ + K_i^-} \tag{17}$$

$$f\left(K_i^+\right) = \frac{K_i^-}{K_i^+ + K_i^-} \tag{18}$$

Step 7: Ranking the alternatives.

## 3. A New Fuzzy MARCOS Method

Step 1: Creating an initial fuzzy decision-making matrix. MCDM models include the definition of a set of n criteria and m alternatives.

Step 2: Creating an *extended* initial fuzzy matrix. The extension is performed by determining the fuzzy anti-ideal $\tilde{A}(AI)$ and fuzzy ideal $\tilde{A}(ID)$ solution.

$$\tilde{X} = \begin{array}{c} \tilde{A}(AI) \\ \tilde{A}_1 \\ \tilde{A}_2 \\ \ldots \\ \tilde{A}_m \\ \tilde{A}(ID) \end{array} \begin{array}{cccc} \tilde{C}_1 & \tilde{C}_2 & \ldots & \tilde{C}_n \\ \left[ \begin{array}{cccc} \tilde{x}_{ai1} & \tilde{x}_{ai2} & \ldots & \tilde{x}_{ain} \\ \tilde{x}_{11} & \tilde{x}_{12} & \ldots & \tilde{x}_{1n} \\ \tilde{x}_{21} & \tilde{x}_{22} & \ldots & \tilde{x}_{2n} \\ \ldots & \ldots & \ldots & \ldots \\ \tilde{x}_{m1} & \tilde{x}_{22} & \ldots & \tilde{x}_{mn} \\ \tilde{x}_{id1} & \tilde{x}_{id2} & \ldots & \tilde{x}_{idn} \end{array} \right] \end{array} \tag{19}$$

The fuzzy $\tilde{A}(AI)$ is the worst alternative while the fuzzy $\tilde{A}(ID)$ is an alternative with the best performance. Depending on type of the criteria, $\tilde{A}(AI)$ and $\tilde{A}(ID)$ are defined by applying Equations (20) and (21):

$$\tilde{A}(AI) = \min_i \tilde{x}_{ij} \; if \; j \in \; B \; and \; \max_i \tilde{x}_{ij} \; if \; j \in C \tag{20}$$

$$\tilde{A}(ID) = \max_i \tilde{x}_{ij} \; if \; j \in \; B \; and \; \min_i \tilde{x}_{ij} \; if \; j \in C \tag{21}$$

*B* belongs to the maximization group of criteria while *C* belongs to the minimization group of criteria.

Step 3: Creating a normalized fuzzy matrix $\tilde{N} = \left[\tilde{n}_{ij}\right]_{m \times n}$ obtained by applying Equations (22) and (23):

$$\tilde{n}_{ij} = \left(n_{ij}^l, n_{ij}^m, n_{ij}^u\right) = \left(\frac{x_{id}^l}{x_{ij}^u}, \frac{x_{id}^l}{x_{ij}^m}, \frac{x_{id}^l}{x_{ij}^l}\right) if \; j \in \; C \tag{22}$$

$$\tilde{n}_{ij} = \left(n_{ij}^l, n_{ij}^m, n_{ij}^u\right) = \left(\frac{x_{ij}^l}{x_{id}^u}, \frac{x_{ij}^m}{x_{id}^u}, \frac{x_{ij}^u}{x_{id}^u}\right) if \; j \in \; B \tag{23}$$

where elements $x_{ij}^l, x_{ij}^m, x_{ij}^u$ and $x_{id}^l, x_{id}^m, x_{id}^u$ represent the elements of the matrix $\tilde{X}$.

Step 4: Computation of the weighted fuzzy matrix $\tilde{V} = \left[\tilde{v}_{ij}\right]_{m \times n}$ Matrix $\tilde{V}$ is calculated by multiplying matrix $\tilde{N}$ with the fuzzy weight coefficients of the criterion $\tilde{w}_j$, Equation (24).

$$\tilde{v}_{ij} = \left(v_{ij}^l, v_{ij}^m, v_{ij}^u\right) = \tilde{n}_{ij} \otimes \tilde{w}_j = \left(n_{ij}^l \times w_j^l, n_{ij}^m \times w_j^m, n_{ij}^u \times w_j^u\right) \tag{24}$$

Step 5: Calculation of $\tilde{S}_i$ fuzzy matrix using the following Equation (25):

$$\tilde{S}_i = \sum_{i=1}^{n} \tilde{v}_{ij} \tag{25}$$

where $\tilde{S}_i\left(s_i^l, s_i^m, s_i^u\right)$ represents the sum of the elements of the weighted fuzzy matrix $\tilde{V}$.

*Step 6*: Calculation of the utility degree of alternatives $\tilde{K}_i$ by applying Equations (26) and (27).

$$\tilde{K}_i^- = \frac{\tilde{S}_i}{\tilde{S}_{ai}} = \left(\frac{s_i^l}{s_{ai}^u}, \frac{s_i^m}{s_{ai}^m}, \frac{s_i^u}{s_{ai}^l}\right) \tag{26}$$

$$\tilde{K}_i^+ = \frac{\tilde{S}_i}{\tilde{S}_{id}} = \left(\frac{s_i^l}{s_{id}^u}, \frac{s_i^m}{s_{id}^m}, \frac{s_i^u}{s_{id}^l}\right) \tag{27}$$

Step 7: Calculation of fuzzy matrix $\tilde{T}_i$ using Equation (28)

$$\tilde{T}_i = \tilde{t}_i = \left(t_i^l, t_i^m, t_i^u\right) = \tilde{K}_i^- \oplus \tilde{K}_i^+ = \left(k_i^{-l} + k_i^{+l}, k_i^{-m} + k_i^{+m}, k_i^{-u} + k_i^{+u}\right) \tag{28}$$

Then, it is necessary to determine a new fuzzy number $\tilde{D}$ using Equation (29)

$$\tilde{D} = \left(d^l, d^m, d^u\right) = \max_i \tilde{t}_{ij} \tag{29}$$

and then, it is necessary to de-fuzzify the number $\tilde{D}$ by using the expression $df_{crisp} = \frac{l+4m+u}{6}$ obtaining the number $df_{crisp}$.

Step 8. Determination of utility functions in relation to the ideal $f\left(\tilde{K}_i^+\right)$ and anti-ideal $f\left(\tilde{K}_i^-\right)$ solution by applying Equations (30) and (31).

$$f\left(\tilde{K}_i^+\right) = \frac{\tilde{K}_i^-}{df_{crisp}} = \left(\frac{k_i^{-l}}{df_{crisp}}, \frac{k_i^{-m}}{df_{crisp}}, \frac{k_i^{-u}}{df_{crisp}}\right) \tag{30}$$

$$f\left(\tilde{K}_i^-\right) = \frac{\tilde{K}_i^+}{df_{crisp}} = \left(\frac{k_i^{+l}}{df_{crisp}}, \frac{k_i^{+m}}{df_{crisp}}, \frac{k_i^{+u}}{df_{crisp}}\right) \tag{31}$$

After that, it is necessary to perform defuzzification for $\tilde{K}_i^-$, $\tilde{K}_i^+$, $f\left(\tilde{K}_i^+\right)$, $f\left(\tilde{K}_i^-\right)$ and apply the following step:

Step 9: Determination of the utility function of alternatives $fK_i$ by Equation (32).

$$f(K_i) = \frac{K_i^+ + K_i^-}{1 + \frac{1-f\left(K_i^+\right)}{f\left(K_i^+\right)} + \frac{1-f\left(K_i^-\right)}{f\left(K_i^-\right)}}; \tag{32}$$

Step 10: Ranking the alternatives based on the final values of utility functions. It is desirable that an alternative have the highest possible value of the utility function.

In addition to developing the new fuzzy MARCOS method, a new linguistic scale for evaluating alternatives has been defined, which is shown in Table 1. A total of nine linguistic terms are defined and for each term, its triangular fuzzy number.

**Table 1.** A newly defined scale for evaluating potential solutions.

| Linguistic Term | Mark | TFN |
|---|---|---|
| Extremely poor | EP | (1,1,1) |
| Very poor | VP | (1,1,3) |
| Poor | P | (1,3,3) |
| Medium poor | MP | (3,3,5) |
| Medium | M | (3,5,5) |
| Medium good | MG | (5,5,7) |
| Good | G | (5,7,7) |
| Very good | VG | (7,7,9) |
| Extremely good | EG | (7,9,9) |

## 4. Results

In order to determine the degree of risk on the roads in Bosnia and Herzegovina through the Rudanka-Doboj M-17 section (length 7.4 km), a list of six criteria was formed on the basis of which the evaluation was carried out. The analysis was conducted in 38 Sections of two-lane main roads of the first order in Bosnia and Herzegovina as potential alternatives. The following are the starting points for the potential criteria affecting traffic risk: a longitudinal gradient (upgrade/downgrade)—(C1), the number of access points on each section (left and right) (C2), the number of traffic accidents with fatalities (C3), the number of traffic accidents with slightly injured persons (C4), the number of traffic accidents with seriously injured persons (C5) and the number of traffic accidents with material damage (C6). The number of traffic accidents for all four classes was taken from the sample for the last four years.

As mentioned above, the Rudanka-Doboj M-17 section covers a total length of 7.4 km and was divided into 200-m sections that represent alternatives. After forming the MCDM model with 38 alternatives and six criteria in the first step, the fuzzy anti-ideal $\tilde{A}(AI)$ and fuzzy ideal $\tilde{A}(ID)$ solutions are defined in the second step on the basis of Equations (20) and (21). Thus, an extended fuzzy initial decision matrix is formed. Table 2 shows the extended initial fuzzy decision matrix with linguistic ratings, as well as quantified values in triangular fuzzy numbers.

**Table 2.** Extended initial fuzzy decision matrix.

| | Linguistic Ratings | | | | | | Ratings with TFNs | | | | | |
|---|---|---|---|---|---|---|---|---|---|---|---|---|
| | C1 | C2 | C3 | C4 | C5 | C6 | C1 | C2 | C3 | C4 | C5 | C6 |
| A1 | EP | VP | EP | M | M | MP | (1,1,1) | (1,1,3) | (1,1,1) | (3,5,5) | (3,5,5) | (3,3,5) |
| A2 | VP | VP | EP | EP | P | VP | (1,1,3) | (1,1,3) | (1,1,1) | (1,1,1) | (1,3,3) | (1,1,3) |
| A3 | MP | VP | EP | EP | EP | VP | (3,3,5) | (1,1,3) | (1,1,1) | (1,1,1) | (1,1,1) | (1,1,3) |
| A4 | M | EP | EP | M | EP | VP | (3,5,5) | (1,1,1) | (1,1,1) | (3,5,5) | (1,1,1) | (1,1,3) |
| A5 | VP | VP | EG | EP | EP | EP | (1,1,3) | (1,1,3) | (7,9,9) | (1,1,1) | (1,1,1) | (1,1,1) |
| A6 | MP | EP | EP | VP | P | VP | (3,3,5) | (1,1,1) | (1,1,1) | (1,1,3) | (1,3,3) | (1,1,3) |
| A7 | P | VP | EP | MP | EP | MG | (1,3,3) | (1,1,3) | (1,1,1) | (3,3,5) | (1,1,1) | (5,5,7) |
| A8 | MG | VP | EP | EP | P | P | (5,5,7) | (1,1,3) | (1,1,1) | (1,1,1) | (1,3,3) | (1,3,3) |
| A9 | EP | VP | EP | EP | G | VP | (1,1,1) | (1,1,3) | (1,1,1) | (1,1,1) | (5,7,7) | (1,1,3) |
| A10 | G | VP | EP | EP | EP | P | (5,7,7) | (1,1,3) | (1,1,1) | (1,1,1) | (1,1,1) | (1,3,3) |
| A11 | VP | VP | EP | EP | EP | VP | (1,1,3) | (1,1,3) | (1,1,1) | (1,1,1) | (1,1,1) | (1,1,3) |
| A12 | M | VP | EG | EP | EP | EP | (3,5,5) | (1,1,3) | (7,9,9) | (1,1,1) | (1,1,1) | (1,1,1) |
| A13 | M | P | EP | VP | EP | VP | (3,5,5) | (1,3,3) | (1,1,1) | (1,1,3) | (1,1,1) | (1,1,3) |
| A14 | MP | M | EP | VP | EP | P | (3,3,5) | (3,5,5) | (1,1,1) | (1,1,3) | (1,1,1) | (1,3,3) |
| A15 | VP | MP | EP | VP | EP | P | (1,1,3) | (3,3,5) | (1,1,1) | (1,1,3) | (1,1,1) | (1,3,3) |
| A16 | VP | VG | EP | VP | EP | EP | (1,1,3) | (7,7,9) | (1,1,1) | (1,1,3) | (1,1,1) | (1,1,1) |
| A17 | MP | EG | EP | P | EP | P | (3,3,5) | (7,9,9) | (1,1,1) | (1,3,3) | (1,1,1) | (1,3,3) |
| A18 | MP | VG | EG | EP | EP | P | (3,3,5) | (7,7,9) | (7,9,9) | (1,1,1) | (1,1,1) | (1,3,3) |
| A19 | MP | G | EP | MP | EP | MP | (3,3,5) | (5,7,7) | (1,1,1) | (3,3,5) | (1,1,1) | (3,3,5) |
| A20 | VP | VG | EP | MG | P | EG | (1,1,3) | (7,7,9) | (1,1,1) | (5,5,7) | (1,3,3) | (7,9,9) |
| A21 | EG | G | EP | EP | P | VP | (7,9,9) | (5,7,7) | (1,1,1) | (1,1,1) | (1,3,3) | (1,1,3) |
| A22 | MG | MP | EP | VP | EP | EP | (5,5,7) | (3,3,5) | (1,1,1) | (1,1,3) | (1,1,1) | (1,1,1) |

**Table 2.** *Cont.*

| | Linguistic Ratings | | | | | | Ratings with TFNs | | | | | |
|---|---|---|---|---|---|---|---|---|---|---|---|---|
| | C1 | C2 | C3 | C4 | C5 | C6 | C1 | C2 | C3 | C4 | C5 | C6 |
| A23 | MP | M | EG | VG | M | EG | (3,3,5) | (3,5,5) | (7,9,9) | (7,7,9) | (3,5,5) | (7,9,9) |
| A24 | VP | MP | EP | EP | EP | VP | (1,1,3) | (3,3,5) | (1,1,1) | (1,1,1) | (1,1,1) | (1,1,3) |
| A25 | P | M | EP | VP | EP | EP | (1,3,3) | (3,5,5) | (1,1,1) | (1,1,3) | (1,1,1) | (1,1,1) |
| A26 | MG | M | EP | EP | EP | VP | (5,5,7) | (3,5,5) | (1,1,1) | (1,1,1) | (1,1,1) | (1,1,3) |
| A27 | M | P | EP | VP | EP | VP | (3,5,5) | (1,3,3) | (1,1,1) | (1,1,3) | (1,1,1) | (1,1,3) |
| A28 | VP | EP | EP | G | M | MP | (1,1,3) | (1,1,1) | (1,1,1) | (5,7,7) | (3,5,5) | (3,3,5) |
| A29 | P | VP | EP | VP | EP | EP | (1,3,3) | (1,1,3) | (1,1,1) | (1,1,3) | (1,1,1) | (1,1,1) |
| A30 | P | VP | EP | VP | EP | VP | (1,3,3) | (1,1,3) | (1,1,1) | (1,1,3) | (1,1,1) | (1,1,3) |
| A31 | EP | VP | EP | VP | EP | P | (1,1,1) | (1,1,3) | (1,1,1) | (1,1,3) | (1,1,1) | (1,3,3) |
| A32 | VP | P | EP | M | P | VP | (1,1,3) | (1,3,3) | (1,1,1) | (3,5,5) | (1,3,3) | (1,1,3) |
| A33 | P | P | EG | VP | EP | MG | (1,3,3) | (1,3,3) | (7,9,9) | (1,1,3) | (1,1,1) | (5,5,7) |
| A34 | VP | EP | EP | EP | EP | VP | (1,1,3) | (1,1,1) | (1,1,1) | (1,1,1) | (1,1,1) | (1,1,3) |
| A35 | P | EP | EP | EP | EP | EP | (1,3,3) | (1,1,1) | (1,1,1) | (1,1,1) | (1,1,1) | (1,1,1) |
| A36 | VP | VP | EP | VP | P | VP | (1,1,3) | (1,1,3) | (1,1,1) | (1,1,3) | (1,3,3) | (1,1,3) |
| A37 | MG | VP | EP | EP | P | MP | (5,5,7) | (1,1,3) | (1,1,1) | (1,1,1) | (1,3,3) | (3,3,5) |
| A38 | MP | VP | EP | EP | EP | VP | (3,3,5) | (1,1,3) | (1,1,1) | (1,1,1) | (1,1,1) | (1,1,3) |

The scope of values of the observed road sections according to each individual criterion are as follows. For the first criterion, the scope of the values ranges from 0% to 1.4%, which generally represents favorable topographic conditions. For the second criterion relating to the total number of access points, the values range from zero to 23. Considering that there are over 20 access points on particular sections, a potential danger to traffic participants and an impact on traffic flow complexity can be noticed. When considering alternatives in relation to the third criterion, it is important to note that there are a total of five sections, with one fatal accident each. For traffic accidents with minor traffic injuries, the values are in the range of 0–7, traffic accidents with serious injuries in the range of 0–3 and material damage in the range of 0–19.

After forming the fuzzy initial decision matrix, it is necessary to determine the significance of input parameters, i.e., their values. For this purpose, the fuzzy PIPRECIA method developed in [20] was applied. As this is an already exploited method, detailed procedures for calculating the values of criteria are not shown, but rather, the summarized results by each step (Table 3).

**Table 3.** Calculation and results of applying the fuzzy PIPRECIA method for determining the criterion values.

| | sj | kj | qj | wj | DF |
|---|---|---|---|---|---|
| C1 | | (1,1,1) | (1,1,1) | (0.136,0.182,0.223) | 0.181 |
| C2 | (1.2,1.3,1.35) | (0.65,0.7,0.8) | (1.25,1.429,1.538) | (0.17,0.261,0.343) | 0.259 |
| C3 | (0.5,0.667,1) | (1,1.333,1.5) | (0.833,1.071,1.538) | (0.113,0.195,0.343) | 0.206 |
| C4 | (0.333,0.4,0.5) | (1.5,1.6,1.667) | (0.5,0.67,1.026) | (0.068,0.122,0.229) | 0.131 |
| C5 | (1.1,1.15,1.2) | (0.8,0.85,0.9) | (0.556,0.788,1.282) | (0.076,0.144,0.286) | 0.156 |
| C6 | (0.4,0.5,0.667) | (1.333,1.5,1.6) | (0.347,0.525,0.962) | (0.047,0.096,0.214) | 0.107 |
| SUM | | | (4.486,5.483,7.346) | | |
| | sj | kj | qj | wj | DF |
| C1 | (0.4,0.5,0.667) | (1.333,1.5,1.6) | (0.827,1.528,2.885) | (0.06,0.165,0.449) | 0.195 |
| C2 | (1.1,1.15,1.2) | (0.8,0.85,0.9) | (1.323,2.292,3.846) | (0.095,0.247,0.599) | 0.281 |
| C3 | (1.3,1.45,1.5) | (0.5,0.55,0.7) | (1.19,1.948,3.077) | (0.086,0.21,0.479) | 0.234 |
| C4 | (0.5,0.667,1) | (1,1.333,1.5) | (0.833,1.071,1.538) | (0.06,0.116,0.24) | 0.127 |
| C5 | (1.2,1.3,1.35) | (0.65,0.7,0.8) | (1.25,1.429,1.538) | (0.09,0.154,0.24) | 0.158 |
| C6 | | (1,1,1) | (1,1,1) | (0.072,0.108,0.156) | 0.110 |
| SUM | | | (6.423,9.268,13.885) | | |

Based on the aggregation of the values *wj* shown in Table 3, the final criterion values are obtained: $\tilde{w}_1 = (0.098, 0.174, 0.336)$, $\tilde{w}_2 = (0.133, 0.254, 0.471)$, $\tilde{w}_3 = (0.100, 0.203, 0.411)$, $\tilde{w}_4 = (0.064, 0.119, 0.234)$, $\tilde{w}_5 = (0.083, 0.149, 0.263)$, $\tilde{w}_6 = (0.060, 0.102, 0.185)$.

The elements of the fuzzy normalized matrix (Table 4) were obtained by applying Equation (23) since all the criteria are of benefit type, i.e., they need to be maximized. An example of normalization is

$$\tilde{n}_{11} = \left(\tfrac{1.000}{9.000}, \tfrac{1.000}{9.000}, \tfrac{1.000}{9.000}\right) = (0.111, 0.111, 0.111),$$
$$\tilde{n}_{15} = \left(\tfrac{3.000}{7.000}, \tfrac{5.000}{7.000}, \tfrac{5.000}{7.000}\right) = (0.429, 0.714, 0.714)$$

**Table 4.** Fuzzy normalized decision matrix.

|     | C1 | C2 | C3 | C4 | C5 | C6 |
|-----|----|----|----|----|----|----|
| AAI | (0.111,0.111,0.111) | (0.111,0.111,0.111) | (0.111,0.111,0.111) | (0.111,0.111,0.111) | (0.143,0.143,0.143) | (0.111,0.111,0.111) |
| A1  | (0.111,0.111,0.111) | (0.111,0.111,0.333) | (0.111,0.111,0.111) | (0.333,0.556,0.556) | (0.429,0.714,0.714) | (0.333,0.333,0.556) |
| A2  | (0.111,0.111,0.333) | (0.111,0.111,0.333) | (0.111,0.111,0.111) | (0.111,0.111,0.111) | (0.143,0.429,0.429) | (0.111,0.111,0.333) |
| A3  | (0.333,0.333,0.556) | (0.111,0.111,0.333) | (0.111,0.111,0.111) | (0.111,0.111,0.111) | (0.143,0.143,0.143) | (0.111,0.111,0.333) |
|     |    |    | ... |    |    |    |
| A23 | (0.333,0.333,0.556) | (0.333,0.556,0.556) | (0.778,1,1) | (0.778,0.778,1) | (0.429,0.714,0.714) | (0.778,1,1) |
| A24 | (0.111,0.111,0.333) | (0.333,0.333,0.556) | (0.111,0.111,0.111) | (0.111,0.111,0.111) | (0.143,0.143,0.143) | (0.111,0.111,0.333) |
| A25 | (0.111,0.333,0.333) | (0.333,0.556,0.556) | (0.111,0.111,0.111) | (0.111,0.111,0.333) | (0.143,0.143,0.143) | (0.111,0.111,0.111) |
|     |    |    | ... |    |    |    |
| A36 | (0.111,0.111,0.333) | (0.111,0.111,0.333) | (0.111,0.111,0.111) | (0.111,0.111,0.333) | (0.143,0.429,0.429) | (0.111,0.111,0.333) |
| A37 | (0.556,0.556,0.778) | (0.111,0.111,0.333) | (0.111,0.111,0.111) | (0.111,0.111,0.111) | (0.143,0.429,0.429) | (0.333,0.333,0.556) |
| A38 | (0.333,0.333,0.556) | (0.111,0.111,0.333) | (0.111,0.111,0.111) | (0.111,0.111,0.111) | (0.143,0.143,0.143) | (0.111,0.111,0.333) |
| ID  | (0.778,1,1) | (0.778,1,1) | (0.778,1,1) | (0.778,0.778,1) | (0.714,1,1) | (0.778,1,1) |

The values of the weighted normalized matrix shown in Table 5 are obtained using Equation (24): $\tilde{v}_{11} = \left(n_{11}^l \times w_1^l, n_{11}^m \times w_1^m, n_{11}^u \times w_1^u\right) = (0.111 \times 0.098, 0.111 \times 0.174, 0.111 \times 0.336) = (0.011, 0.019, 0.037)$.

**Table 5.** Fuzzy weighted normalized decision matrix.

|     | C1 | C2 | C3 | C4 | C5 | C6 |
|-----|----|----|----|----|----|----|
| AAI | (0.011,0.019,0.037) | (0.015,0.028,0.052) | (0.011,0.023,0.046) | (0.007,0.013,0.026) | (0.012,0.021,0.038) | (0.007,0.011,0.021) |
| A1  | (0.011,0.019,0.037) | (0.015,0.028,0.157) | (0.011,0.023,0.046) | (0.021,0.066,0.13) | (0.035,0.106,0.188) | (0.02,0.034,0.103) |
| A2  | (0.011,0.019,0.112) | (0.015,0.028,0.157) | (0.011,0.023,0.046) | (0.007,0.013,0.026) | (0.012,0.064,0.113) | (0.007,0.011,0.062) |
| A3  | (0.033,0.058,0.187) | (0.015,0.028,0.157) | (0.011,0.023,0.046) | (0.007,0.013,0.026) | (0.012,0.021,0.038) | (0.007,0.011,0.062) |
|     |    |    | ... |    |    |    |
| A23 | (0.033,0.058,0.187) | (0.044,0.141,0.262) | (0.077,0.203,0.411) | (0.05,0.092,0.234) | (0.035,0.106,0.188) | (0.046,0.102,0.185) |
| A24 | (0.011,0.019,0.112) | (0.044,0.085,0.262) | (0.011,0.023,0.046) | (0.007,0.013,0.026) | (0.012,0.021,0.038) | (0.007,0.011,0.062) |
| A25 | (0.011,0.058,0.112) | (0.044,0.141,0.262) | (0.011,0.023,0.046) | (0.007,0.013,0.078) | (0.012,0.021,0.038) | (0.007,0.011,0.021) |
|     |    |    | ... |    |    |    |
| A36 | (0.011,0.019,0.112) | (0.015,0.028,0.157) | (0.011,0.023,0.046) | (0.007,0.013,0.078) | (0.012,0.064,0.113) | (0.007,0.011,0.062) |
| A37 | (0.054,0.096,0.261) | (0.015,0.028,0.157) | (0.011,0.023,0.046) | (0.007,0.013,0.026) | (0.012,0.064,0.113) | (0.02,0.034,0.103) |
| A38 | (0.033,0.058,0.187) | (0.015,0.028,0.157) | (0.011,0.023,0.046) | (0.007,0.013,0.026) | (0.012,0.021,0.038) | (0.007,0.011,0.062) |
| ID  | (0.076,0.174,0.336) | (0.103,0.254,0.471) | (0.077,0.203,0.411) | (0.05,0.092,0.234) | (0.059,0.149,0.263) | (0.046,0.102,0.185) |

The fuzzy matrix $\tilde{S}_i$ is obtained by applying Equation (25)

$$
\begin{aligned}
&\tilde{S}_{ai} = (0.062, 0.116, 0.219) &&\tilde{S}\ldots = , (\ldots, \ldots, \ldots) &&\tilde{S}_{,36} = (0.062, 0.158, 0.567) \\
&\tilde{S}_1 = (0.113, 0.276, 0.660) &&\tilde{S}_{23} = (0.286, 0.702, 1.466) &&\tilde{S}_{37} = (0.119, 0.258, 0.705) \\
&\tilde{S}_2 = (0.062, 0.158, 0.515) \quad ' &&\tilde{S}_{24} = (0.092, 0.172, 0.544) \quad ' &&\tilde{S}_{38} = (0.084, 0.154, 0.514) \\
&\tilde{S}_3 = (0.084, 0.154, 0.514) &&\tilde{S}_{25} = (0.092, 0.267, 0.555) &&\tilde{S}_{ID} = (0.412, 0.974, 1.900)
\end{aligned}
$$

as follows:

$$\tilde{S}_1 = \begin{pmatrix} 0.111 + 0.015 + 0.011 + 0.021 + 0.035 + 0.020, \\ 0.019 + 0.028 + 0.023 + 0.066 + 0.106 + 0.034, \\ 0.037 + 0.157 + 0.046 + 0.130 + 0.188 + 0.103 \end{pmatrix} = (0.113, 0.276, 0.660)$$

Using Equation (26), the matrix $\tilde{K}_i^-$ is obtained:

$$\begin{array}{lll}
\tilde{k}_1^- = (0.517, 2.386, 10.607) & \tilde{k}_{23}^- = (1.304, 6.064, 23.546) & \tilde{k}_{36}^- = (0.284, 1.367, 9.105) \\
\tilde{k}_2^- = (0.284, 1.367, 8.270) \quad , & \tilde{k}_{24}^- = (0.418, 1.487, 8.745) \quad , & \tilde{k}_{37}^- = (0.542, 2.229, 11.329) \\
\tilde{k}_3^- = (0.383, 1.333, 8.264) & \tilde{k}_{25}^- = (0.418, 2.307, 8.920) & \tilde{k}_{38}^- = (0.383, 1.333, 8.264)
\end{array}$$

as follows:

$$\tilde{k}_1^- = \frac{\tilde{S}_1}{\tilde{S}_{ai}} = \left( \frac{s_1^l}{s_{ai}^u}, \frac{s_1^m}{s_{ai}^m}, \frac{s_1^u}{s_{ai}^l} \right) = \left( \frac{0.113}{0.219}, \frac{0.276}{0.116}, \frac{0.660}{0.062} \right) = (0.517, 2.386, 10.607)$$

Using Equation (27), the matrix $\tilde{K}_i^+$ is obtained:

$$\begin{array}{lll}
\tilde{k}_1^+ = (0.060, 0.284, 1.602) & \tilde{k}_{23}^+ = (0.151, 0.721, 3.557) & \tilde{k}_{36}^+ = (0.033, 0.163, 1.375) \\
\tilde{k}_2^+ = (0.033, 0.163, 1.249) \quad , & \tilde{k}_{24}^+ = (0.048, 0.177, 1.321) \quad , & \tilde{k}_{37}^+ = (0.063, 0.265, 1.711) \\
\tilde{k}_3^+ = (0.044, 0.159, 1.248) & \tilde{k}_{25}^+ = (0.048, 0.275, 1.348) & \tilde{k}_{38}^+ = (0.044, 0.159, 1.248)
\end{array}$$

as follows:

$$\tilde{k}_1^+ = \frac{\tilde{S}_1}{\tilde{S}_{id}} = \left( \frac{s_1^l}{s_{id}^u}, \frac{s_1^m}{s_{id}^m}, \frac{s_1^u}{s_{id}^l} \right) = \left( \frac{0.113}{1.900}, \frac{0.276}{0.974}, \frac{0.660}{0.412} \right) = (0.060, 0.284, 1.602)$$

In Step 7, the matrix $\tilde{T}_i$ is calculated using Equation (28):

$$\begin{array}{l}
\tilde{t}_1 = (0.577, 2.670, 12.210), \; \tilde{t}_{23} = (1.454, 6.785, 27.103), \; \tilde{t}_{36} = (0.317, 1.530, 10.481), \\
\tilde{t}_2 = (0.317, 1.530, 9.519), \; \tilde{t}_{24} = (0.466, 1.664, 10.066), \; \tilde{t}_{37} = (0.605, 2.494, 13.040), \\
\tilde{t}_3 = (0.427, 1.492, 9.512), \; \tilde{t}_{25} = (0.466, 2.582, 10.268), \; \tilde{t}_{38} = (0.427, 1.492, 9.512),
\end{array}$$

The elements of the matrix $\tilde{T}_i$ are obtained as follows:

$$\tilde{t}_1 = (0.517 + 0.060, 2.386 + 0.284, 10.607 + 1.602) = (0.577, 2.670, 12.210)$$

Then, it is necessary to determine a new fuzzy number $\tilde{D}$ using Equation (29) $\tilde{D} = \left( d^l, d^m, d^u \right) = \max_i \tilde{t}_{ij}$ and $\tilde{D} = (1.454, 6.785, 27.103)$ is obtained, and then it is necessary to defuzzify the number $\tilde{D}$ by using the expression $df_{crisp} = \frac{l + 4m + u}{6}$ obtaining the number $df_{crisp} = 9.283$. The calculation of the last two steps and the final results obtained using the fuzzy MARCOS method are shown in Table 6.

**Table 6.** Calculation of the two last steps and results of applied fuzzy MARCOS.

| | $f(\tilde{K}_i^-)$ | $f(\tilde{K}_i^+)$ | K- | K+ | fK- | fK+ | Ki | Rank |
|---|---|---|---|---|---|---|---|---|
| A1 | (0.006,0.031,0.173) | (0.056,0.257,1.143) | 3.445 | 0.466 | 0.050 | 0.371 | 0.181 | 15 |
| A2 | (0.004,0.018,0.135) | (0.031,0.147,0.891) | 2.337 | 0.322 | 0.035 | 0.252 | 0.084 | 29 |
| A3 | (0.005,0.017,0.134) | (0.041,0.144,0.89) | 2.330 | 0.321 | 0.035 | 0.251 | 0.083 | 30 |
| | | .... | | | | | | |
| A23 | (0.016,0.078,0.383) | (0.14,0.653,2.536) | 8.184 | 1.099 | 0.118 | 0.882 | 1.082 | 1 |
| A24 | (0.005,0.019,0.142) | (0.045,0.16,0.942) | 2.519 | 0.346 | 0.037 | 0.271 | 0.097 | 27 |
| A25 | (0.005,0.03,0.145) | (0.045,0.249,0.961) | 3.095 | 0.416 | 0.045 | 0.333 | 0.144 | 20 |
| | | ... | | | | | | |
| A36 | (0.004,0.018,0.148) | (0.031,0.147,0.981) | 2.476 | 0.343 | 0.037 | 0.267 | 0.095 | 28 |
| A37 | (0.007,0.029,0.184) | (0.058,0.24,1.22) | 3.464 | 0.472 | 0.051 | 0.373 | 0.185 | 14 |
| A38 | (0.005,0.017,0.134) | (0.041,0.144,0.89) | 2.330 | 0.321 | 0.035 | 0.251 | 0.083 | 30 |

Utility functions in relation to the ideal $f\left(\tilde{K}_i^+\right)$ and anti-ideal $f\left(\tilde{K}_i^-\right)$ solution are determined by applying Equations (30) and (31).

$$f\left(\tilde{K}_1^+\right) = \frac{\tilde{K}_1^-}{df_{crisp}} = \left(\frac{0.517}{9.283}, \frac{2.386}{9.283}, \frac{10.607}{9.283}\right)$$

$$f\left(\tilde{K}_1^-\right) = \frac{\tilde{K}_1^+}{df_{crisp}} = \left(\frac{0.060}{9.283}, \frac{0.284}{9.283}, \frac{1.602}{9.283}\right)$$

After that, it is necessary to perform defuzzification for $\tilde{K}_i^-$, $\tilde{K}_i^+$, $f\left(\tilde{K}_i^+\right)$, $f\left(\tilde{K}_i^-\right)$, which is given in Table 6.

Determination of the utility function of alternatives $fK_i$. The utility function of alternatives is defined by Equation (32).

$$f(K_1) = \frac{K_1^+ + K_1^-}{1 + \frac{1-f\left(K_1^+\right)}{f\left(K_1^+\right)} + \frac{1-f\left(K_1^-\right)}{f\left(K_1^-\right)}} = \frac{0.446 + 3.445}{1 + \frac{1-0.371}{0.371} + \frac{1-0.050}{0.050}} = 0.181$$

The ranking represents the sorting of obtained values in descending order, where A23 represents the most hazardous 200-m section and is dominant over the others.

## 5. Validation Tests

### 5.1. Changing the Significance of Input Parameters

In the first phase of validation test, the impact of changing the three most significant criteria C1, C2 and C3 on ranking results was analyzed. Using Equation (33), a total of 30 scenarios were created.

$$\tilde{W}_{n\beta} = \left(1 - \tilde{W}_{n\alpha}\right) \frac{\tilde{W}_\beta}{\left(1 - \tilde{W}_n\right)} \tag{33}$$

The scenarios were formed through three different groups of 10 sets each. In the first group of scenarios, the first criterion was changed, criterion C2 was changed in the second group and criterion C3 was changed in the third group. If Equation (33) is observed, $\tilde{W}_{n\beta}$ represents the fuzzy corrected value of the criteria C2, C3, C4, C5 and C6, then C1, C3, C4, C5 and C6, i.e., C1, C2, C4, C5 and C6, respectively by groups. $\tilde{W}_{n\alpha}$ represents the reduced fuzzy value of the criteria C1, C2, and C3 respectively by groups, $\tilde{W}_\beta$ represents the original fuzzy value of the criterion considered and $\tilde{W}_n$ represents the original fuzzy value of the criterion whose value is reduced, in this case, C1, C2 and C3.

In the first scenario, the fuzzy value of criterion C1 was reduced by 5% while the values of the remaining criteria were proportionally corrected by applying Equation (33). In each subsequent scenario, the value of criterion C1 was reduced by 10%, while the values of the remaining criteria were corrected, so they met the condition $\sum_{j=1}^{n} w_j^m = 1$. These changes, i.e., these 10 scenarios represent the first group. Scenarios 11–20 represent the second group in which criterion C2 was corrected. The third group consists of scenarios 21–30 with the change in the value of the third criterion. After forming 30 new vectors of the weight coefficients of the criteria (Table 7), new model results were obtained, as presented in Figures 1–3.

**Table 7.** New criterion values across 30 scenarios.

|  | w1 | w2 | w3 | w4 | w5 | w6 |
|---|---|---|---|---|---|---|
| S1 | (0.093,0.165,0.319) | (0.133,0.257,0.483) | (0.1,0.205,0.421) | (0.064,0.12,0.24) | (0.083,0.15,0.269) | (0.06,0.103,0.19) |
| S2 | (0.083,0.148,0.286) | (0.135,0.262,0.507) | (0.101,0.209,0.442) | (0.065,0.123,0.252) | (0.084,0.154,0.283) | (0.061,0.105,0.199) |
| S3 | (0.073,0.13,0.252) | (0.136,0.267,0.53) | (0.102,0.213,0.463) | (0.066,0.125,0.264) | (0.085,0.157,0.296) | (0.061,0.107,0.208) |
| S4 | (0.064,0.113,0.218) | (0.138,0.273,0.554) | (0.103,0.218,0.484) | (0.066,0.128,0.276) | (0.086,0.16,0.309) | (0.062,0.109,0.218) |
| S5 | (0.054,0.095,0.185) | (0.139,0.278,0.578) | (0.104,0.222,0.505) | (0.067,0.13,0.287) | (0.087,0.163,0.322) | (0.063,0.111,0.227) |
| S6 | (0.044,0.078,0.151) | (0.141,0.283,0.602) | (0.106,0.226,0.525) | (0.068,0.133,0.299) | (0.088,0.166,0.336) | (0.063,0.114,0.237) |
| S7 | (0.034,0.061,0.118) | (0.142,0.289,0.626) | (0.107,0.231,0.546) | (0.069,0.135,0.311) | (0.089,0.169,0.349) | (0.064,0.116,0.246) |
| S8 | (0.024,0.043,0.084) | (0.144,0.294,0.65) | (0.108,0.235,0.567) | (0.069,0.138,0.323) | (0.09,0.172,0.362) | (0.064,0.118,0.255) |
| S9 | (0.015,0.026,0.05) | (0.145,0.299,0.673) | (0.109,0.239,0.588) | (0.07,0.14,0.335) | (0.09,0.176,0.376) | (0.065,0.12,0.265) |
| S10 | (0.005,0.009,0.017) | (0.146,0.305,0.697) | (0.11,0.243,0.609) | (0.071,0.143,0.347) | (0.091,0.179,0.389) | (0.066,0.122,0.274) |
| S11 | (0.099,0.177,0.351) | (0.126,0.241,0.447) | (0.1,0.206,0.429) | (0.065,0.121,0.244) | (0.083,0.151,0.274) | (0.06,0.104,0.193) |
| S12 | (0.1,0.182,0.381) | (0.113,0.216,0.4) | (0.102,0.213,0.466) | (0.066,0.125,0.265) | (0.085,0.157,0.298) | (0.061,0.107,0.21) |
| S13 | (0.102,0.188,0.411) | (0.1,0.19,0.353) | (0.103,0.22,0.502) | (0.066,0.129,0.286) | (0.086,0.162,0.321) | (0.062,0.111,0.226) |
| S14 | (0.103,0.194,0.441) | (0.086,0.165,0.306) | (0.105,0.227,0.539) | (0.067,0.133,0.307) | (0.087,0.167,0.344) | (0.063,0.114,0.243) |
| S15 | (0.105,0.2,0.471) | (0.073,0.14,0.259) | (0.106,0.234,0.576) | (0.068,0.137,0.328) | (0.089,0.172,0.368) | (0.064,0.117,0.259) |
| S16 | (0.106,0.206,0.5) | (0.06,0.114,0.212) | (0.108,0.241,0.612) | (0.069,0.141,0.349) | (0.09,0.177,0.391) | (0.065,0.121,0.276) |
| S17 | (0.108,0.212,0.53) | (0.046,0.089,0.165) | (0.109,0.248,0.649) | (0.07,0.145,0.369) | (0.091,0.182,0.415) | (0.066,0.124,0.292) |
| S18 | (0.109,0.218,0.56) | (0.033,0.063,0.118) | (0.111,0.255,0.685) | (0.071,0.149,0.39) | (0.092,0.187,0.438) | (0.066,0.128,0.308) |
| S19 | (0.111,0.224,0.59) | (0.02,0.038,0.071) | (0.113,0.261,0.722) | (0.072,0.153,0.411) | (0.094,0.192,0.461) | (0.067,0.131,0.325) |
| S20 | (0.112,0.23,0.62) | (0.007,0.013,0.024) | (0.114,0.268,0.758) | (0.073,0.157,0.432) | (0.095,0.197,0.485) | (0.068,0.135,0.341) |
| S21 | (0.098,0.176,0.348) | (0.133,0.257,0.487) | (0.095,0.193,0.39) | (0.064,0.12,0.242) | (0.083,0.151,0.272) | (0.06,0.103,0.191) |
| S22 | (0.099,0.18,0.371) | (0.135,0.264,0.52) | (0.085,0.172,0.349) | (0.065,0.123,0.259) | (0.084,0.155,0.29) | (0.061,0.106,0.204) |
| S23 | (0.101,0.185,0.395) | (0.136,0.27,0.553) | (0.075,0.152,0.308) | (0.066,0.126,0.275) | (0.085,0.158,0.308) | (0.061,0.108,0.217) |
| S24 | (0.102,0.189,0.418) | (0.138,0.277,0.586) | (0.065,0.132,0.267) | (0.067,0.129,0.291) | (0.086,0.162,0.327) | (0.062,0.111,0.23) |
| S25 | (0.103,0.194,0.441) | (0.139,0.283,0.619) | (0.055,0.112,0.226) | (0.067,0.132,0.308) | (0.087,0.166,0.345) | (0.063,0.114,0.243) |
| S26 | (0.104,0.198,0.465) | (0.141,0.289,0.652) | (0.045,0.091,0.185) | (0.068,0.136,0.324) | (0.088,0.17,0.363) | (0.063,0.116,0.256) |
| S27 | (0.105,0.202,0.488) | (0.142,0.296,0.684) | (0.035,0.071,0.144) | (0.069,0.139,0.34) | (0.089,0.174,0.382) | (0.064,0.119,0.269) |
| S28 | (0.106,0.207,0.512) | (0.144,0.302,0.717) | (0.025,0.051,0.103) | (0.069,0.142,0.357) | (0.09,0.177,0.4) | (0.065,0.121,0.282) |
| S29 | (0.107,0.211,0.535) | (0.145,0.309,0.75) | (0.015,0.03,0.062) | (0.07,0.145,0.373) | (0.091,0.181,0.418) | (0.065,0.124,0.295) |
| S30 | (0.108,0.216,0.559) | (0.147,0.315,0.783) | (0.005,0.01,0.021) | (0.071,0.148,0.389) | (0.092,0.185,0.437) | (0.066,0.126,0.308) |

After forming scenarios as described above, results were obtained for each of the groups. Figure 2 shows the obtained ranks of the alternatives and comparison of initial results with the results in the first group of scenarios, i.e., S1–S10.

The change in the significance of the first criterion affects the ranks of road network sections, which is, in a way, understandable since there is a large number of alternatives. It is important to note that alternatives A23, A18, A20, A12, A5 and A16, which take the first, second, third, eighth, ninth and tenth places, respectively, do not change their positions in any scenario. With the elimination of the significance of the first criterion, since its value is 0.05 in the tenth scenario, there is a change in ranks by five positions for some alternatives.

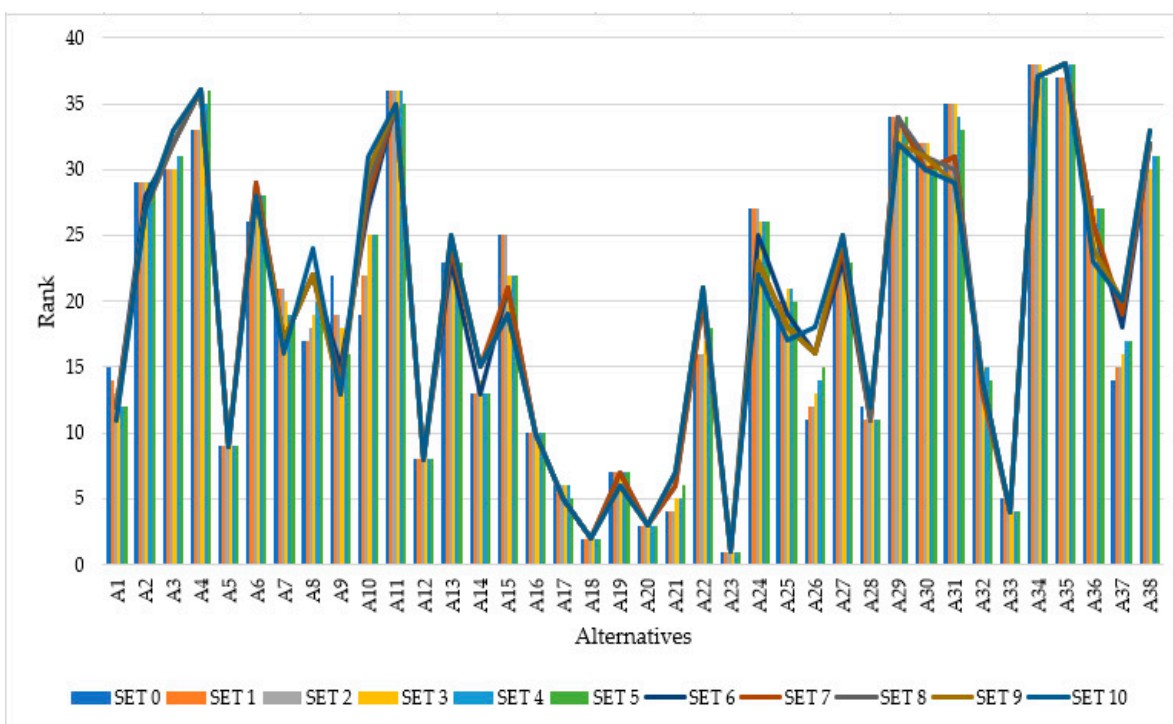

**Figure 2.** Comparison of initial results with scenarios S1–S10.

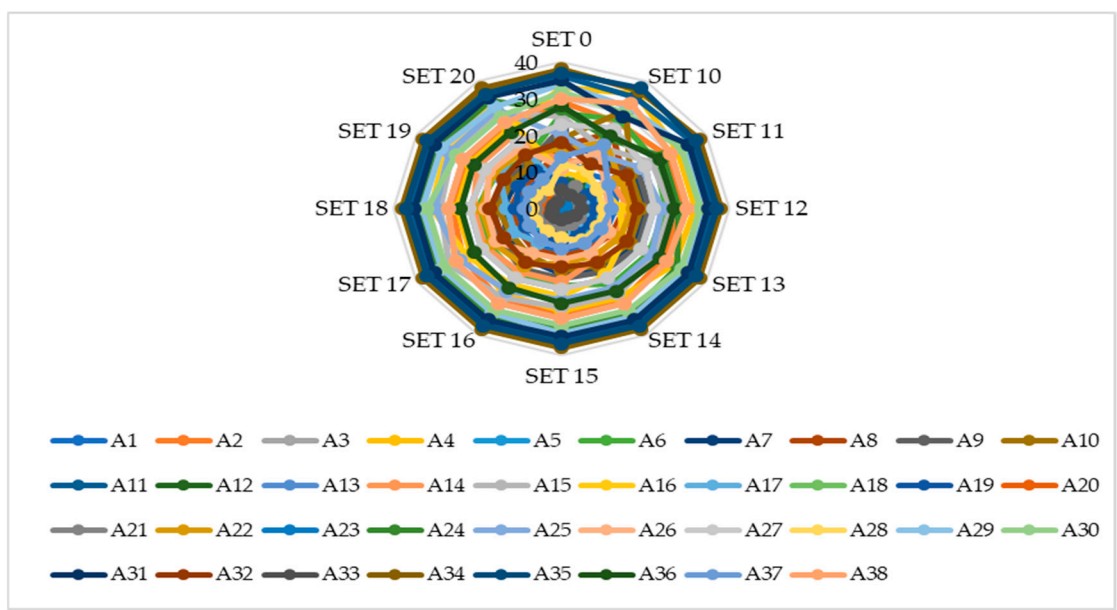

**Figure 3.** Comparison of initial results with the second group of scenarios, i.e., scenarios S11–S20.

Figure 3 shows the obtained ranks of the alternatives and comparison of initial results with the results in the second group of scenarios, i.e., S11–S20. In Figure 3, it can be noticed that there have been major changes in the ranks of alternatives across scenarios. The reason for this is the fact that the most significant criterion C2 has been changed, which has a significant impact on the output. However, alternative A23 remains in the first place, despite the fact that the influence of the most significant criterion is minimized. Additionally, alternatives A31 and A34 do not change their position and are ranked 35th and 38th, respectively. Alternative A18 in scenarios S11–S17 retains its second position, while in the remaining three scenarios (S18–S20), it is in the third place. Alternative A20 is in the third

position in scenarios S11–S13, fourth in S14-S16 and fifth in S17–S20 scenarios. With the decrease in the impact of criterion C2, results and ranks change to a maximum of 37%.

Figure 4 shows the obtained ranks of the alternatives and a comparison of initial results with the results in the third group of scenarios, i.e., S21–S30.

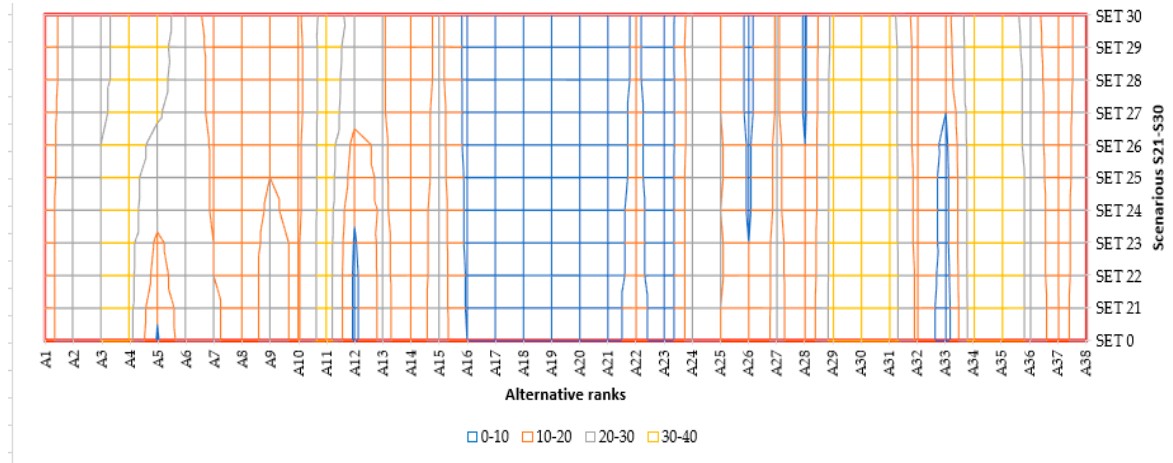

**Figure 4.** Comparison of initial results with third group of scenarios i.e., S21–S30.

In Figure 4, it can be noticed that there were also some changes in the ranks of the alternatives across the scenarios. The reason for this is the fact that the second most significant criterion C3 was changed, which has a slightly lower impact than C2, but is also very important in obtaining the output. However, alternatives A23, A34 and A35 do not change their ranks and they are still ranked as first, 38th and 37th, respectively. Alternative A20 retains its third position in scenarios S21 and S22, while holding second place in the remaining scenarios. Alternative A30 changes its position by one place only in the last S30 scenario.

### 5.2. Impact of Reverse Rank Matrices

One of the ways to test the validity of the obtained results of the model for decision-making is to construct dynamic matrices and then analyze the solutions that the model provides under newly formed conditions. If the solutions show some logical contradictions that are expressed in the form of undesirable changes in the ranks of alternatives, then one may express concern that there is a problem with the mathematical apparatus of the applied method. In line with this goal, a test in which the resistance of the model to the rank reversal problem is considered was conducted.

A change in the number of alternatives was made for each scenario, eliminating the worst alternative from further consideration. After defining a new set of alternatives, the ranking of the remaining alternatives is performed under newly formed conditions using the proposed model. In the test, 35 scenarios were formed in which the change in the elements of the decision matrix was simulated. As a rule, 37 scenarios should be formed (one less than the total number of alternatives). However, in this case, we had two newly formed scenarios in which two alternatives were eliminated. In scenario S7, there are two alternatives A3 and A38 that have the same position, and both alternatives were eliminated in the next eighth scenario. The same situation is with scenario S14 in which A13 and A27 were eliminated.

Based on the results obtained (Figure 5) and taking into account the complexity of the MCDM model, it can be concluded that the model is stable. In the first 14 scenarios, there is no change in ranks for any alternative. Only eight alternatives change their ranks after the fifteenth scenario has been formed when Alternative A9 is eliminated from the model. It is important to note that after obtaining the results in S15 when the change occurs, the alternatives retain their ranks until the last scenario.

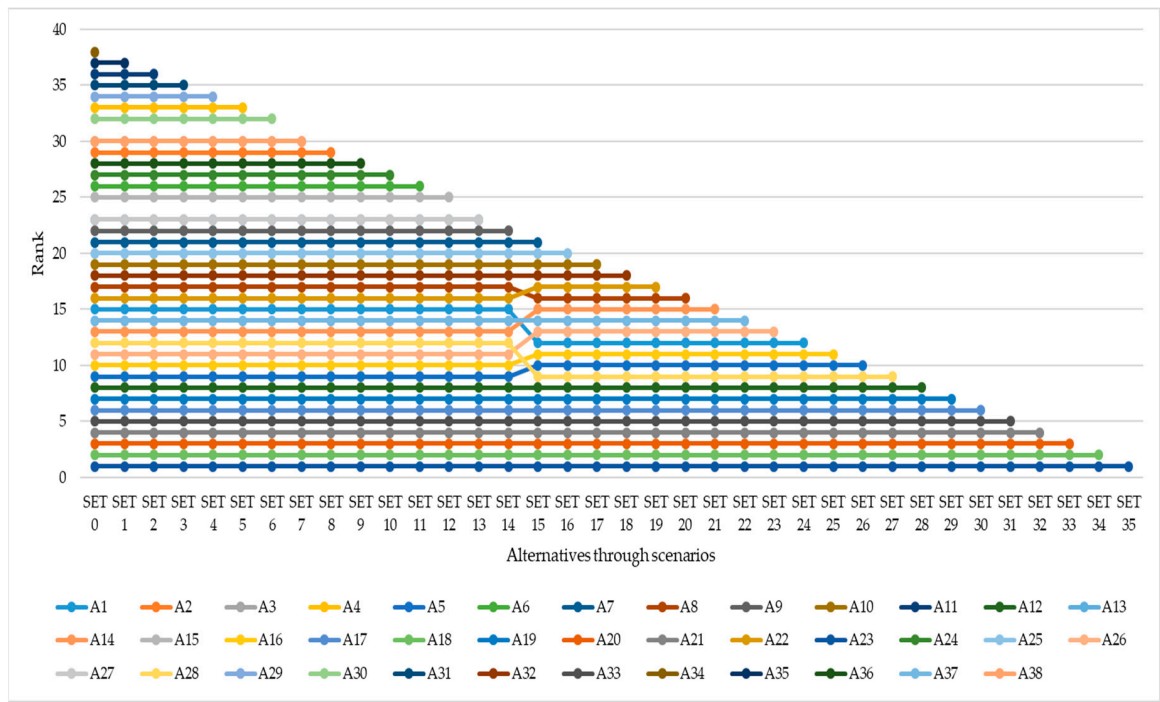

**Figure 5.** Results of the test of reverse rank matrix.

*5.3. Comparison with Other Approaches*

In this section, a validation test is performed involving comparison with two other methods in a fuzzy form: the Fuzzy SAW [21] and the fuzzy TOPSIS method [22], and the results are presented in Figure 6.

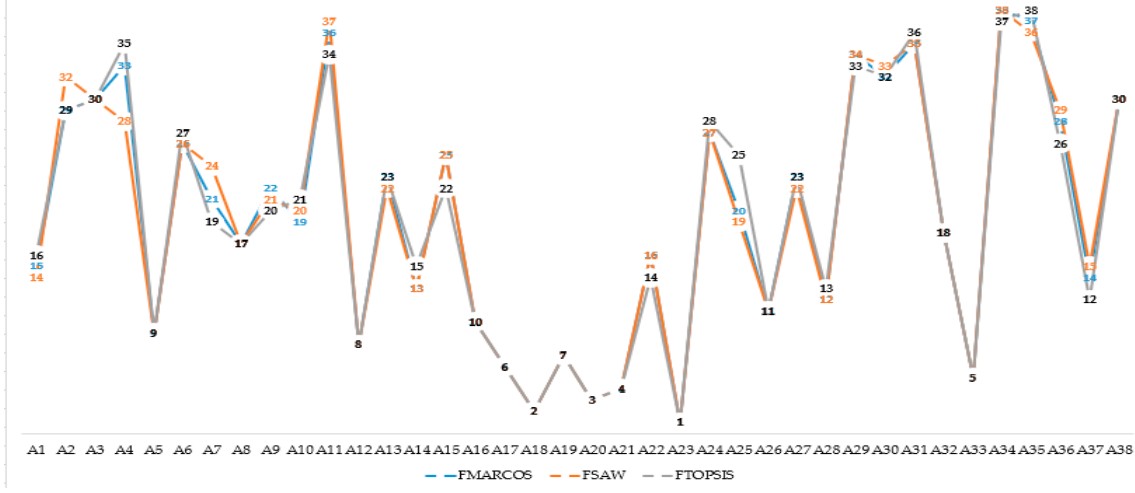

**Figure 6.** Results of comparison with Fuzzy SAW and fuzzy TOPSIS methods.

In this case study, we compared the fuzzy MARCOS technique with MCDM models that have a linear normalization. When observing the comparisons of the applied methods, it can be seen that in 15 cases, there is no change of ranks. It is of primary importance to note that the first 11 (A23, A18, A20, A21, A33, A17, A19, A12, A5, A16, and A26) alternatives do not change positions regardless of the approach taken. In addition, alternatives A8, A32, A3 and A38 do not change their ranks and are in 17th, 18th and 30th places, respectively. The last two mentioned alternatives share the 30th position. In addition, compared to the fuzzy SAW method, eight other alternatives do not change their

ranks, while compared to the fuzzy TOPSIS method, three other alternatives do not change positions. The biggest difference is with alternative A25, which is ranked 20th applying the fuzzy MARCOS method and in the 25th position using the fuzzy TOPSIS method. The observed small differences in ranking of alternatives, between Fuzzy MARCOS results and Fuzzy SAW/Fuzzy TOPSIS results (see Figure 6) do not limit the usefulness of the study, as it is impossible to know in advance the possible outcomes of an applied methodology and the extent of possible deviation of the rankings, therefore making the process is significant for validation.

The advantages of the Fuzzy MARCOS method are as follows: consideration of fuzzy reference points through the fuzzy ideal and fuzzy anti-ideal solution at the very beginning of model formation, more precise determination of the degree of utility with respect to both set solutions, proposal of a new way of determining utility functions and its aggregation, possibility to consider a large set of criteria and alternatives, as demonstrated through a realistic example, too. Compared with other methods, this method is simple, effective, and easy to sort and optimize the process.

## 6. Conclusions

In this paper, a fuzzy MARCOS algorithm was developed to support multi-criteria decision-making, especially when considering parameters in an uncertain environment. Considering the relationships of indicators presented through TFNs between the ideal and the anti-ideal solution, it can positively affect making valid decisions. In addition, this paper defines a new fuzzy linguistic scale for parameter evaluation by decision-makers. The Fuzzy MARCOS model was tested using the example of determining the degree of risk on short sections of the first-order main road. A part of the road network with a length of 7.4 km divided into 38 short sections of 200 m each was analyzed. Thanks to the previously formed adequate database regarding all necessary parameters, the MCDM model was created. The results of the proposed model show that the 23rd section, i.e., the section between 4.2 and 4.4 km represents the most hazardous section since the value obtained is drastically higher than the others are. This result is caused by the fact that this section has an undesirable value considering almost all factors and an adequate reaction in terms of increasing surveillance and traffic safety on this section is required. The obtained results in terms of risk can be used for improving road safety. The results can help decision-makers take into account these indicators as an input parameter for all planning, design and operational analyzes, as well as indicators for the development of regulatory plans for a given area in local conditions. For the purpose of validation, an extensive analysis was carried out, which involves changing the significance of the input parameters, testing the factors of dynamic environment and comparing the results with two other methods in a fuzzy form. The validation tests support the development and application of the fuzzy MARCOS method. In order to improve the robustness of MCDM in fuzzy environment, a new fuzzy MARCOS method was developed in this study, which uses the ratio method and the reference point method to obtain a scheme of basic comprehensive decision information. The fuzzy MARCOS method is a powerful tool for optimizing multiple goals. Fuzzy MARCOS refreshes the MCDM domain by introducing an algorithm for analyzing the relationship between alternatives and reference points. The Fuzzy MARCOS method integrates the following points to provide a robust decision: defining reference points (fuzzy ideal and fuzzy anti-ideal values), determining the relationship between alternatives and fuzzy ideal/anti-ideal values, defining the utility degree of alternatives in relation to the fuzzy ideal and fuzzy anti-ideal solutions. The results obtained by the fuzzy MARCOS method are more reasonable due to the fusion of the results of the ratio approach and reference point sorting approach. The Fuzzy MARCOS method shows the significant stability and reliability of the results in a dynamic environment. Moreover, it is important to note that in numerous scenarios, the fuzzy MARCOS method shows stability in processing large data sets, which was proven in the performed research.

Future research may be based on the integration of particular short sections and their evaluation, and the development of the MARCOS method with other theories such as neutrophic [23], single-valued intuitionistic fuzzy numbers [24], grey theory [25] and others. Moreover, the approach of a building

consensus in group decision making with information granularity [26] or the concept of a granular fuzzy preference relation where each pairwise comparison is formed as a certain information granule can be implemented [27].

**Author Contributions:** Conceptualization, M.S. (Miomir Stanković) and M.S. (Marko Subotić); methodology, Ž.S.; validation, D.K.D. and D.P.; investigation, M.S. (Marko Subotić); writing—original draft preparation, Ž.S.; writing—review and editing, D.K.D. and D.P.; supervision, M.S. (Miomir Stanković). All authors have read and agreed to the published version of the manuscript.

**Funding:** This research received no external funding.

**Acknowledgments:** The paper is a part of the research done within the project No. 19.032/961-58/19 "Influence of Geometric Elements of Two-lane Roads in Traffic Risk Analysis Models" supported by Ministry of Scientific and Technological Development, Higher Education and Information Society of the Republic of Srpska.

**Conflicts of Interest:** The authors declare no conflict of interest.

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
