# Peer review of "A New Fuzzy MARCOS Method for Road Traffic Risk Analysis"

_mathematics, doi:10.3390/math8030457_

Round 1

Reviewer 1 Report

Review of A New Fuzzy MARCOS Method for Road Traffic Risk Analysis

I am quite satisfied with this paper. The most interesting part of the paper is its experimentation, which is really extensive. Authors have put lot of efforts for properly show the advantadges of their proposal with the experimentation, and they properly explain the significance of their results. 

Anyway, authors have missed several important aspects in the rest of the paper. Authors should clearly explain which is the main motivation of their work, and how this motivation has led to the main contribution of the paper, which should also be clearly detailed. Furthermore, authors should describe the main originalities of the work. Related to this, I highly recommend authors to add a new version explaining the current state of the art of the topic. Authors should elaborate very hard on this sense, in order to properly place their work in the current community.

Author Response

Reviewer 1:

Thank you very much for the useful suggestions. We accepted all of the suggestions and we are sure that his will improve the quality and contribute to a better understanding of the paper.

I am quite satisfied with this paper. The most interesting part of the paper is its experimentation, which is really extensive. Authors have put lot of efforts for properly show the advantadges of their proposal with the experimentation, and they properly explain the significance of their results.

Comment: Anyway, authors have missed several important aspects in the rest of the paper. Authors should clearly explain which is the main motivation of their work, and how this motivation has led to the main contribution of the paper, which should also be clearly detailed. Furthermore, authors should describe the main originalities of the work. Related to this, I highly recommend authors to add a new version explaining the current state of the art of the topic. Authors should elaborate very hard on this sense, in order to properly place their work in the current community.

Response to comment: Thank you for your comment. The main motivation and contributions are added in the introduction section. Also, their connection is presented. The main originalities of the paper have presented in the conclusion section.

Reviewer 2 Report

This paper presents a practical application of a MCDM called MARCOS.In my opinion, the significance of the results is good enouhg to be published in this journal. However, there are some issues to improve in order to reach the quelity of the journal.

Structure of the paper is good but english language should be improved. Introduction section could also be improved by including a deeper justification of the paper and different MCDM references as:

J.A. Morente-Molinera, G. Kou, I.J. Pérez, K. Samuylov, A. Selamat, E. Herrera-Viedma. A group decision making support system for the Web: how to work in environments with a high number of participants and alternatives. Applied Soft Computing, 68 (2018) 191-201.

Section 3 presents a 10 steps method. The inclusion of a flow diagram could be appropriate for the reader understanding.

Figure 5 shows a very similar results when comparing the proposed method with SAW and TOPSIS. A deeper discussion on why and when or where MARCOS is better than others is required to give significance to authors results.

As a suggestion, maybe in the future, authors could improve their research by using granular computing:

F.J. Cabrerizo, J.A. Morente-Molinera, W. Pedrycz, A. Taghavi, E. Herrera-Viedma. Granulating linguistic information in decision making under consensus and consistency. Expert Systems With Applications 99 (2018) 83-92.
F.J. Cabrerizo, M.R. Ureña, W. Pedrycz, E. Herrera-Viedma. Building consensus in group decision making with an allocation of information granularity. Fuzzy Sets and Systems 255 (2014) 115-127

Author Response

Reviewer 2:

Thank you very much for the useful suggestions. We accepted all of the suggestions and we are sure that his will improve the quality and contribute to a better understanding of the paper.

This paper presents a practical application of a MCDM called MARCOS.In my opinion, the significance of the results is good enouhg to be published in this journal. However, there are some issues to improve in order to reach the quelity of the journal.

Comment 1: Structure of the paper is good but english language should be improved. Introduction section could also be improved by including a deeper justification of the paper and different MCDM references as:

J.A. Morente-Molinera, G. Kou, I.J. Pérez, K. Samuylov, A. Selamat, E. Herrera-Viedma. A group decision making support system for the Web: how to work in environments with a high number of participants and alternatives. Applied Soft Computing, 68 (2018) 191-201.

Response to comment 1: We have improved the paper. We have cited the mentioned reference and also extended the introduction section according to your suggestion.

Comment 2: Section 3 presents a 10 steps method. The inclusion of a flow diagram could be appropriate for the reader understanding.

Response to comment 2: Thank you for comment. We have added algorithms of the new developed fuzzy MARCOS method as Figure 1. We have shown all the steps in the short line.

Comment 3: Figure 5 shows a very similar results when comparing the proposed method with SAW and TOPSIS. A deeper discussion on why and when or where MARCOS is better than others is required to give significance to authors results.

Response to comment 3: We have added advantages of developed fuzzy MARCOS method as follow: The advantages of the Fuzzy MARCOS method are as follows: consideration of fuzzy reference points through the fuzzy ideal and fuzzy anti-ideal solution at the very beginning of model formation, more precise determination of the degree of utility with respect to both set solutions, proposal of a new way of determining utility functions and its aggregation, possibility to consider a large set of criteria and alternatives, as demonstrated through a realistic example, too. Compared with other methods, this method is simple, effective, and easy to sort and optimize the process.

Comment 4: As a suggestion, maybe in the future, authors could improve their research by using granular computing:

F.J. Cabrerizo, J.A. Morente-Molinera, W. Pedrycz, A. Taghavi, E. Herrera-Viedma. Granulating linguistic information in decision making under consensus and consistency. Expert Systems With Applications 99 (2018) 83-92.

F.J. Cabrerizo, M.R. Ureña, W. Pedrycz, E. Herrera-Viedma. Building consensus in group decision making with an allocation of information granularity. Fuzzy Sets and Systems 255 (2014) 115-127

Response to comment 4: Thank you for your suggestion. We have added this in conclusion as part of future research: Also, can be implemented the approach of building consensus in group decision making with information granularity [24] or concept of a granular fuzzy preference relation where each pairwise comparison is formed as a certain information granule [25].

Reviewer 3 Report

General comments

The manuscript is proposing a new fuzzy multi-criteria decision-making model for traffic risk assessment, in which an outranking methodology is combined with fuzzy logic for estimating the risk in section of a road network.

While the research is generally well written (e.g. structure, introduction, methodology, results), the results are hardly different from existing, well known MCDA methods (such as TOPSIS) and the manuscript possibly does not bring novelty or significance in the sector. The analysis of results is quite poor, describing the rankings of alternative scenarios but without providing any information on how the proposed MCDA analysis can be used to improve the formulated problem or how decision makers can take advantage of the results or which is the future practical value the analysis brings.

Specific comments per section

  1. Introduction
  • Line 57: Please use a consistent citing methodology, as for [8] and [10]. Thus, cite properly study [11], e.g. ‘Bao et al., [11] used a fuzzy methodology for similar purposes.’

The same applies to other citations in the manuscript, so I would suggest revising the whole manuscript accordingly.

2 Preliminaries

  • Line 87-88: Please use consistent terminology. Ideal (and anti-ideal) solution in Lines 84-85 and Ideal (anti-ideal) alternative in these lines.
  • Line 86 and 90: Only Cost criteria (C) are denoted in the decision matrix (C1, C2, …, Cn), there is no reference to Benefit criteria (B), which are then presented in line 90.
  • I would strongly encourage revising the denotation of terms in section 2, as I observe inconsistencies in their use (C, B, m/n and i/j, nij, etc)
  • Line 108-109: Text should be “using Equations (17) and (18)”, instead of (11) and (12) as written in the manuscript

3 A new fuzzy MARCOS method

  • Line 116-117: Again, inconsistent denotations and terminology, in comparison to section 2.
  • Line 158: Can scholars justify the definition of the linguistic scale and how the terms have been selected? Why EG is represented by (7,9,9) and not (9,9,9)? Similarly, why M is (3,5,5) and not (5,5,5)?

4 Results

  • Line 201-202: A Risk assessment can easily include analysis with cost criteria, therefore the sentence “they need to be maximized as the objective function is risk assessment” is wrong. Please rephrase accordingly

5.1 Changing the significance of input parameters

  • Line 265: Figure lacks information on scales, axis titles, legend etc. and is generally unreadable.
  • Line 273: should be “second group of scenarios, i.e. S11-S20”
  • Line 275: Comment on Figure 1 applies to Figure 2. All three Figures (1-2-3) are in practice unreadable
  • Figures 1-2-3: Scholars use three different types of figures to present the same type of information. Why is that?
  • Line 286: should be “third group of scenarios, i.e. S21-S30”

5.2 Impact of reverse rank matrices

  • Line 313: Figure lacks information on scales, axis titles, legend etc.

5.3 Comparison with other approaches

  • There are two ways to look at the ‘stability’ of the proposed model through the validation with other approaches. Why do we consider a new approach necessary when two already existing approaches would be sufficient for analyzing the given MCDM problem?

6 Conclusion

  • Scholars should provide information on the utility of the analysis results to decision makers or how they can be used to help improving the problem under analysis.

Author Response

Reviewer 3:

Thank you very much for the suggestions. We accepted some of the suggestions.

Comment 1: The manuscript is proposing a new fuzzy multi-criteria decision-making model for traffic risk assessment, in which an outranking methodology is combined with fuzzy logic for estimating the risk in section of a road network.

While the research is generally well written (e.g. structure, introduction, methodology, results), the results are hardly different from existing, well known MCDA methods (such as TOPSIS) and the manuscript possibly does not bring novelty or significance in the sector. The analysis of results is quite poor, describing the rankings of alternative scenarios but without providing any information on how the proposed MCDA analysis can be used to improve the formulated problem or how decision makers can take advantage of the results or which is the future practical value the analysis brings.

Response to comment 1: We think that contributions of this paper can’t be neglected since we have developed new MCDM method which is validated thorough four independent reviewers in prestige SCI indexed journal. In this paper we have developed fuzzy extension of said method and showed benefits of it’s application on real world engineering problem. Furthermore, the case study is very complex, since the MCMD problem has large number of alternatives (which is rare). We gave all calculation steps in detail and proven that our model is good. We have used various tests validation and gave advantages and contributions of our new fuzzy MARCOS method. So, such negative comment is subjective and general without facts and detail arguments.  Also, the quality of our model is confirmed by comments of other anonymous reviewers.

Specific comments per section

Comment 2: 1. Introduction

  • Line 57: Please use a consistent citing methodology, as for [8] and [10]. Thus, cite properly study [11], e.g. ‘Bao et al., [11] used a fuzzy methodology for similar purposes.’

The same applies to other citations in the manuscript, so I would suggest revising the whole manuscript accordingly.

Response to comment 2: We have used consistent citing methodology in the whole paper, where is applicable. In some sentences no need for name of authors, for example above equations (2).

Comment 3: 2 Preliminaries

  • Line 87-88: Please use consistent terminology. Ideal (and anti-ideal) solution in Lines 84-85 and Ideal (anti-ideal) alternative in these lines.

Response to comment 3: Thank you. Corrected.

Comment 4:

  • Line 86 and 90: Only Cost criteria (C) are denoted in the decision matrix (C1, C2, …, Cn), there is no reference to Benefit criteria (B), which are then presented in line 90.

Response to comment 4: The reviewer didn’t understand definition of initial decision matrix which consist as variables alternatives Ai (i=1,2,...,m) and criteria Cj (j=1,2,...,n). In initial decision-making matrix type of criteria hasn’t considered. So, Cj (j=1,2,...,n) denotes both type of criteria (Benefits and Cost type). Please, see equation (7), which represents a common initial matrix with standard elements, except AAI and AI.

Comment 5:

  • I would strongly encourage revising the denotation of terms in section 2, as I observe inconsistencies in their use (C, B, m/n and i/j, nij, etc)

Response to comment 5: We have not understood you. These denotations are standard for such studies.

Comment 6:

  • Line 108-109: Text should be “using Equations (17) and (18)”, instead of (11) and (12) as written in the manuscript

Response to comment 6: Thank you. Corrected.

Comment 7: 3 A new fuzzy MARCOS method

  • Line 116-117: Again, inconsistent denotations and terminology, in comparison to section 2.

Response to comment 7: Terminology must be different. In the second section is presented crisp MARCOS method, while in the third section we have developed fuzzy MARCOS method. Crisp and fuzzy marks must to be different.

Comment 8:

  • Line 158: Can scholars justify the definition of the linguistic scale and how the terms have been selected? Why EG is represented by (7,9,9) and not (9,9,9)? Similarly, why M is (3,5,5) and not (5,5,5)?

Response to comment 8: It seems that the reviewer isn’t familiar with fuzzy theory. Such comment is the result of misunderstandings of fuzzy theory and triangular fuzzy numbers. The basic point of fuzzy number is interval value and membership of each element to the interval. So, if we use the same values for both limits (left and right) of fuzzy number, we don’t have fuzzy number any more. In that case we have crisp number that can’t express any uncertainty in decision making process. For example, if we have  M=(5,5,5) ie. we don’t have fuzzy logic any more. We have crisp number. We can propose to the reviewer some quality papers in fuzzy decision making field.

Comment 9: 4 Results

  • Line 201-202: A Risk assessment can easily include analysis with cost criteria, therefore the sentence “they need to be maximized as the objective function is risk assessment” is wrong. Please rephrase accordingly

Response to comment 9: We have changed this sentence.

Comment 10: 5.1 Changing the significance of input parameters

  • Line 265: Figure lacks information on scales, axis titles, legend etc. and is generally unreadable.

Response to comment 10: We have added axis titles.

Comment 11:

  • Line 273: should be “second group of scenarios, i.e. S11-S20”

Response to comment comment 11: Corrected.

Comment 12:

  • Line 275: Comment on Figure 1 applies to Figure 2. All three Figures (1-2-3) are in practice unreadable

Response to comment comment 12: We have added axis title in Figure 3, while Figure 2 left unchanged.

Comment 13:

  • Figures 1-2-3: Scholars use three different types of figures to present the same type of information. Why is that?

Response to comment comment 13: We have obtained various results depending on each group of scenarios, so we use the most appropriate chart to show comparison results.

Comment 14:

  • Line 286: should be “third group of scenarios, i.e. S21-S30”

Response to comment 14: Corrected.

Comment 15: 5.2 Impact of reverse rank matrices

  • Line 313: Figure lacks information on scales, axis titles, legend etc.

Response to comment 15: Replaced.

Comment 16: 5.3 Comparison with other approaches

  • There are two ways to look at the ‘stability’ of the proposed model through the validation with other approaches. Why do we consider a new approach necessary when two already existing approaches would be sufficient for analyzing the given MCDM problem?

Response to comment 16: It is well known fact that MCDM tools with different type of normalization methods can evaluate the same problem in different way. So, in this case study we have compared fuzzy MARCOS technique with MCDM models that have linear normalization. Furthermore, validation process with two more MCMD techniques gives enough information for showing robustness of proposed results. Because of that we think this is subjective and general comment. Couls you please provide us some details in this direction? We didn’t see in literature any suggestions regarding minimum and maximum number of MCDM tools that should be used for validation. Could you please refer us to literature that suggests such information. Please clearly read advantages of proposed model. We have added this in 342-348 lines.

Comment 17: 6 Conclusion

  • Scholars should provide information on the utility of the analysis results to decision makers or how they can be used to help improving the problem under analysis.

Response to comment 17: We have added new information of proposed methodology in conclusion section.

Round 2

Reviewer 1 Report

Review of A New Fuzzy MARCOS Method for Road Traffic Risk Analysis

Authors have properly addressed all my concerns. For this reason, I suggest the acceptance of this paper.

Author Response

Dear reviewer,

Thank you for your additional review and acceptance of our paper.

Reviewer 3 Report

Dear authors, thank you very much for the effort and improvements presented in the revised version of the manuscript. Please find below some further consideration to my comments and your responses.

General comments

The manuscript is proposing a new fuzzy multi-criteria decision-making model for traffic risk assessment, in which an outranking methodology is combined with fuzzy logic for estimating the risk in section of a road network.

While the research is generally well written (e.g. structure, introduction, methodology, results), the results are hardly different from existing, well known MCDA methods (such as TOPSIS) and the manuscript possibly does not bring novelty or significance in the sector. The analysis of results is quite poor, describing the rankings of alternative scenarios but without providing any information on how the proposed MCDA analysis can be used to improve the formulated problem or how decision makers can take advantage of the results or which is the future practical value the analysis brings.

Response to comment 1: We think that contributions of this paper can’t be neglected since we have developed new MCDM method which is validated thorough four independent reviewers in prestige SCI indexed journal. In this paper we have developed fuzzy extension of said method and showed benefits of it’s application on real world engineering problem. Furthermore, the case study is very complex, since the MCMD problem has large number of alternatives (which is rare). We gave all calculation steps in detail and proven that our model is good. We have used various tests validation and gave advantages and contributions of our new fuzzy MARCOS method. So, such negative comment is subjective and general without facts and detail arguments. Also, the quality of our model is confirmed by comments of other anonymous reviewers.

Response to authors’ comment 1: The general comment is based on the contribution of the study in the sector of 'risk analysis of road traffic' and the way the results of the application of the methodology are presented in the study. We are not evaluating or reviewing the MCDM methodology, which is very interesting and significant, this has been done in a different publication and is out of the scope of the present review process. We are trying to evaluate the significance of the results of the application of the methodology in the sector under analysis. I am sorry that authors see the comment as negative, but it is well known that the MCDM sector includes large uncertainty therefore proper justification on the motivations of a study, on the analysis of the results and how those can be beneficial to science is a prerequisite to high quality research articles. Our role as reviewers is to challenge the authors and trigger discussions if necessary. In this view, it is very important that the introduction, comparison with other approaches and conclusions have been updated.

Specific comments per section

Comment 2: 1. Introduction

  • Line 57: Please use a consistent citing methodology, as for [8] and [10]. Thus, cite properly study [11], e.g. ‘Bao et al., [11] used a fuzzy methodology for similar purposes.’

The same applies to other citations in the manuscript, so I would suggest revising the whole manuscript accordingly.

Response to comment 2: We have used consistent citing methodology in the whole paper, where is applicable. In some sentences no need for name of authors, for example above equations (2).

Response to authors’ comment 2: Thank you

Comment 3: 2 Preliminaries

  • Line 87-88: Please use consistent terminology. Ideal (and anti-ideal) solution in Lines 84-85 and Ideal (anti-ideal) alternative in these lines.

Response to comment 3: Thank you. Corrected.

Response to authors’ comment 3: In the revised manuscript (mathematics-737865-peer-review-v2.pdf) this change has not been incorporated. Please revise.

Comment 4:

  • Line 86 and 90: Only Cost criteria (C) are denoted in the decision matrix (C1, C2, …, Cn), there is no reference to Benefit criteria (B), which are then presented in line 90.

Response to comment 4: The reviewer didn’t understand definition of initial decision matrix which consist as variables alternatives Ai (i=1,2,...,m) and criteria Cj (j=1,2,...,n). In initial decision-making matrix type of criteria hasn’t considered. So, Cj (j=1,2,...,n) denotes both type of criteria (Benefits and Cost type). Please, see equation (7), which represents a common initial matrix with standard elements, except AAI and AI.

Response to authors’ comment 4: The fact that you are denoting Criteria with Cj and Cost type criteria with C creates confusion and inconsistency. I am very familiar with decision matrices, where each element should be denoted properly. This is why my next comment was to revise denotations as a whole in the manuscript.  

Comment 5:

  • I would strongly encourage revising the denotation of terms in section 2, as I observe inconsistencies in their use (C, B, m/n and i/j, nij, etc)

Response to comment 5: We have not understood you. These denotations are standard for such studies.

Response to authors’ comment 5: In relation also to comment 4, denotations should be revised. m is modal value in section 2 and used in the decision matrix for Am. n is used in the definition of Criteria (Cn) and then used to define normalised values in matrix N. In Si (i=1,2,…,m) and equation (15) holds (i=1,…,n). And so on. For these reasons, a revision of equations and denotations is strongly advised.

Comment 6:

  • Line 108-109: Text should be “using Equations (17) and (18)”, instead of (11) and (12) as written in the manuscript

Response to comment 6: Thank you. Corrected.

Response to authors’ comment 6: In Line 118 the revised text still includes “using Equations (12) and (18)”. Please update.

Comment 7: 3 A new fuzzy MARCOS method

  • Line 116-117: Again, inconsistent denotations and terminology, in comparison to section 2.

Response to comment 7: Terminology must be different. In the second section is presented crisp MARCOS method, while in the third section we have developed fuzzy MARCOS method. Crisp and fuzzy marks must to be different.

Response to authors’ comment 7: Thank you for the explanation, obviously crisp and fuzzy marks should be different. Let me give you examples to showcase better what I mean: crisp worst alternative is (AAI) and best alternative is (AI). Fuzzy worst alternative is A~ (AI), which is logical, but best fuzzy alternative is A~ (ID), instead of A~(I) based on the crisp equivalent. Then considerations of Comment 5 also apply to this section.

Comment 8:

  • Line 158: Can scholars justify the definition of the linguistic scale and how the terms have been selected? Why EG is represented by (7,9,9) and not (9,9,9)? Similarly, why M is (3,5,5) and not (5,5,5)?

Response to comment 8: It seems that the reviewer isn’t familiar with fuzzy theory. Such comment is the result of misunderstandings of fuzzy theory and triangular fuzzy numbers. The basic point of fuzzy number is interval value and membership of each element to the interval. So, if we use the same values for both limits (left and right) of fuzzy number, we don’t have fuzzy number any more. In that case we have crisp number that can’t express any uncertainty in decision making process. For example, if we have M=(5,5,5) ie. we don’t have fuzzy logic any more. We have crisp number. We can propose to the reviewer some quality papers in fuzzy decision making field.

Response to authors’ comment 8: Thank you for the clarification, even if it was a trivial issue for you. I am not very familiar with TFN, thus the void comment. Apologies for the misunderstanding.

Comment 9: 4 Results

  • Line 201-202: A Risk assessment can easily include analysis with cost criteria, therefore the sentence “they need to be maximized as the objective function is risk assessment” is wrong. Please rephrase accordingly

Response to comment 9: We have changed this sentence.

Response to authors’ comment 9: Thank you

Comment 10: 5.1 Changing the significance of input parameters

  • Line 265: Figure lacks information on scales, axis titles, legend etc. and is generally unreadable.

Response to comment 10: We have added axis titles.

Response to authors’ comment 10: Thank you

Comment 11:

  • Line 273: should be “second group of scenarios, i.e. S11-S20”

Response to comment 11: Corrected.

Response to authors’ comment 11: Line 281 still says ‘results in the first group of scenarios, i.e. S11 - S20.’ Please update

Comment 12:

  • Line 275: Comment on Figure 1 applies to Figure 2. All three Figures (1-2-3) are in practice unreadable

Response to comment 12: We have added axis title in Figure 3, while Figure 2 left unchanged.

Response to authors’ comment 12: Thank you for the improvement of Figure 3. The general comment remains valid though.

Comment 13:

  • Figures 1-2-3: Scholars use three different types of figures to present the same type of information. Why is that?

Response to comment 13: We have obtained various results depending on each group of scenarios, so we use the most appropriate chart to show comparison results.

Response to authors’ comment 13: The three graphs show the same comparison of ‘rank of alternatives’ towards ‘groups of scenarios’. For comparison reasons you could provide the same type of Figure for each group of scenarios, i.e. Type of Figure 1 for scenario group 2 and scenario group 3 also, as it is the easier to read. And if deemed necessary to provide “other types” of results as mentioned in your response, that could be done with extra charts, case by case for scenario groups 2 and 3.

Comment 14:

  • Line 286: should be “third group of scenarios, i.e. S21-S30”

Response to comment 14: Corrected.

Response to authors’ comment 14: Line 294 still says ‘results in the first group of scenarios, i.e. S21 - S30’. Please update

Comment 15: 5.2 Impact of reverse rank matrices

  • Line 313: Figure lacks information on scales, axis titles, legend etc.

Response to comment 15: Replaced.

Response to authors’ comment 15: Thank you

Comment 16: 5.3 Comparison with other approaches

  • There are two ways to look at the ‘stability’ of the proposed model through the validation with other approaches. Why do we consider a new approach necessary when two already existing approaches would be sufficient for analyzing the given MCDM problem?

Response to comment 16: It is well known fact that MCDM tools with different type of normalization methods can evaluate the same problem in different way. So, in this case study we have compared fuzzy MARCOS technique with MCDM models that have linear normalization. Furthermore, validation process with two more MCMD techniques gives enough information for showing robustness of proposed results. Because of that we think this is subjective and general comment. Couls you please provide us some details in this direction? We didn’t see in literature any suggestions regarding minimum and maximum number of MCDM tools that should be used for validation. Could you please refer us to literature that suggests such information. Please clearly read advantages of proposed model. We have added this in 342-348 lines.

Response to authors’ comment 16: The text above is the type of consideration that you should include in the manuscript section 5.3 and I find text in lines 342-348 as very relevant. Please consider adding also “in this case study we have compared fuzzy MARCOS technique with MCDM models that have linear normalization.” somewhere in the text, as it is important. I would also propose to add that “The observed small differences in ranking of alternatives, between Fuzzy MARCOS results and Fuzzy SAW/Fuzzy TOPSIS results (see Figure 6) do not limit the usefulness of the study, as it is impossible to know in advance the possible outcomes of an applied methodology and the extend of possible deviation of the rankings, therefore making the process significant for validation”.

As mentioned in the introduction of the review, our role is to challenge authors and provide constructive comments for improving the manuscript. The consideration “why a new applied methodology is relevant and important, after the analysis of the results is available” is a question that every researcher in operational research always asks himself/herself because it is very difficult to draw the line between significant improvement/value of a study/importance of case study results. Reviewer 3 has also pointed out this issue, thus lines 342-348 have been added.

Comment 17: 6 Conclusion

  • Scholars should provide information on the utility of the analysis results to decision makers or how they can be used to help improving the problem under analysis.

Response to comment 17: We have added new information of proposed methodology in conclusion section.

Response to authors’ comment 17: Excellent improvement, thank you. In addition, I would suggest describing in few lines how the ranking of alternatives in terms of risk can be used for improving road safety and how the results can be useful for decision makers.

Author Response

Dear reviewer,

Thank you for your suggestions. We additionally improved our paper.

Comment 3: 2 Preliminaries

  • Line 87-88: Please use consistent terminology. Ideal (and anti-ideal) solution in Lines 84-85 and Ideal (anti-ideal) alternative in these lines.

Response to comment 3: Thank you. Corrected.

Response to authors’ comment 3: In the revised manuscript (mathematics-737865-peer-review-v2.pdf) this change has not been incorporated. Please revise.

Response to reviewer: Thank you. We have corrected it. Please see lines 94 and 95.

Comment 4:

  • Line 86 and 90: Only Cost criteria (C) are denoted in the decision matrix (C1, C2, …, Cn), there is no reference to Benefit criteria (B), which are then presented in line 90.

Response to comment 4: The reviewer didn’t understand definition of initial decision matrix which consist as variables alternatives Ai (i=1,2,...,m) and criteria Cj (j=1,2,...,n). In initial decision-making matrix type of criteria hasn’t considered. So, Cj (j=1,2,...,n) denotes both type of criteria (Benefits and Cost type). Please, see equation (7), which represents a common initial matrix with standard elements, except AAI and AI.

Response to authors’ comment 4: The fact that you are denoting Criteria with Cj and Cost type criteria with C creates confusion and inconsistency. I am very familiar with decision matrices, where each element should be denoted properly. This is why my next comment was to revise denotations as a whole in the manuscript. 

Response to reviewer: In all published papers that deal with MCDM problem, criteria are denoted with Cj, and a group of cost criteria with C. This fact is familiar for all readers, even for them which is on the beginning level. Taking this into account we are sure that anyone will not be confused with these marks.

Comment 5:

  • I would strongly encourage revising the denotation of terms in section 2, as I observe inconsistencies in their use (C, B, m/n and i/j, nij, etc)

Response to comment 5: We have not understood you. These denotations are standard for such studies.

Response to authors’ comment 5: In relation also to comment 4, denotations should be revised. m is modal value in section 2 and used in the decision matrix for Amn is used in the definition of Criteria (Cn) and then used to define normalised values in matrix N. In S(i=1,2,…,m) and equation (15) holds (i=1,…,n). And so on. For these reasons, a revision of equations and denotations is strongly advised.

Response to reviewer: This comment is related to our previous reply. Please carefully read it. Also, is obvious that exist difference for marks in index and non-index. For example, standard denotes for criteria are Cn where is clear that n is the number of criteria. Nij denotes normalized matrix (large N). For all readers, the difference is obvious and clear.

Comment 6:

  • Line 108-109: Text should be “using Equations (17) and (18)”, instead of (11) and (12) as written in the manuscript

Response to comment 6: Thank you. Corrected.

Response to authors’ comment 6: In Line 118 the revised text still includes “using Equations (12) and (18)”. Please update.

Response to reviewer: We apologize for the mistake. We have updated this.

Comment 7: 3 A new fuzzy MARCOS method

  • Line 116-117: Again, inconsistent denotations and terminology, in comparison to section 2.

Response to comment 7: Terminology must be different. In the second section is presented crisp MARCOS method, while in the third section we have developed fuzzy MARCOS method. Crisp and fuzzy marks must to be different.

Response to authors’ comment 7: Thank you for the explanation, obviously crisp and fuzzy marks should be different. Let me give you examples to showcase better what I mean: crisp worst alternative is (AAI) and best alternative is (AI). Fuzzy worst alternative is A~ (AI), which is logical, but best fuzzy alternative is A~ (ID), instead of A~(I) based on the crisp equivalent. Then considerations of Comment 5 also apply to this section.

Response to reviewer: We have understood you but isn't rule that marks in crisp and fuzzy theory must be identical. For example, in order to avoid similarity with other published papers, authors in some of their papers are changed marks for previously developed methods which they only have used for computation.

Comment 11:

  • Line 273: should be “second group of scenarios, i.e. S11-S20”

Response to comment comment 11: Corrected.

Response to authors’ comment 11: Line 281 still says ‘results in the first group of scenarios, i.e. S11 - S20.’ Please update

Response to reviewer: We apologize for the mistake. We have updated this. Please see lines 281 and 294.

Comment 13:

  • Figures 1-2-3: Scholars use three different types of figures to present the same type of information. Why is that?

Response to comment comment 13: We have obtained various results depending on each group of scenarios, so we use the most appropriate chart to show comparison results.

Response to authors’ comment 13: The three graphs show the same comparison of ‘rank of alternatives’ towards ‘groups of scenarios’. For comparison reasons you could provide the same type of Figure for each group of scenarios, i.e. Type of Figure 1 for scenario group 2 and scenario group 3 also, as it is the easier to read. And if deemed necessary to provide “other types” of results as mentioned in your response, that could be done with extra charts, case by case for scenario groups 2 and 3.

Response to reviewer: As we have noticed previously we have obtained various results depending on each group of scenarios, so we use the most appropriate chart to show comparison results. Besides, the right of authors is that use charts which they want.

Comment 14:

  • Line 286: should be “third group of scenarios, i.e. S21-S30”

Response to comment 14: Corrected.

Response to authors’ comment 14: Line 294 still says ‘results in the first group of scenarios, i.e. S21 - S30’. Please update.

Response to reviewer: Updated.

Comment 16: 5.3 Comparison with other approaches

  • There are two ways to look at the ‘stability’ of the proposed model through the validation with other approaches. Why do we consider a new approach necessary when two already existing approaches would be sufficient for analyzing the given MCDM problem?

Response to comment 16: It is well known fact that MCDM tools with different type of normalization methods can evaluate the same problem in different way. So, in this case study we have compared fuzzy MARCOS technique with MCDM models that have linear normalization. Furthermore, validation process with two more MCMD techniques gives enough information for showing robustness of proposed results. Because of that we think this is subjective and general comment. Couls you please provide us some details in this direction? We didn’t see in literature any suggestions regarding minimum and maximum number of MCDM tools that should be used for validation. Could you please refer us to literature that suggests such information. Please clearly read advantages of proposed model. We have added this in 342-348 lines.

Response to authors’ comment 16: The text above is the type of consideration that you should include in the manuscript section 5.3 and I find text in lines 342-348 as very relevant. Please consider adding also “in this case study we have compared fuzzy MARCOS technique with MCDM models that have linear normalization.” somewhere in the text, as it is important. I would also propose to add that “The observed small differences in ranking of alternatives, between Fuzzy MARCOS results and Fuzzy SAW/Fuzzy TOPSIS results (see Figure 6) do not limit the usefulness of the study, as it is impossible to know in advance the possible outcomes of an applied methodology and the extend of possible deviation of the rankings, therefore making the process significant for validation”.

Response to reviewer: Thank you for your suggestions. We have added both remarks. See lines 334-335 and 343-347.

Comment 17: 6 Conclusion

  • Scholars should provide information on the utility of the analysis results to decision makers or how they can be used to help improving the problem under analysis.

Response to comment 17: We have added new information of proposed methodology in conclusion section.

Response to authors’ comment 17: Excellent improvement, thank you. In addition, I would suggest describing in few lines how the ranking of alternatives in terms of risk can be used for improving road safety and how the results can be useful for decision makers.

Response to reviewer: We have added the folowing in conclusion: Obtained results in terms of risk can be used for improving road safety. Results can help to decision-makers that these indicators should be an input parameter of all planning, design and operational analyzes, as well as indicators for the development of regulatory plans for a given area in local conditions.

General response. Thanks to all useful comments whole paper have been improved including more contributions and corrections of few mistakes.